# GRAPH SIGNAL SAMPLING FOR INDUCTIVE ONE-BIT MATRIX COMPLETION: A CLOSED-FORM SOLUTION

**Chao Chen**[1], **Haoyu Geng**[1], **Gang Zeng**[2], **Zhaobing Han**[2]
**Hua Chai**[2], **Xiaokang Yang**[1], **Junchi Yan**[1]*
[1]MoE Key Lab of Artificial Intelligence, Shanghai Jiao Tong University
[2]Didi Chuxing
{chao.chen,yanjunchi}@sjtu.edu.cn
Code: https://github.com/cchao0116/GSIMC-ICLR2023

## ABSTRACT

Inductive one-bit matrix completion is motivated by modern applications such as recommender systems, where new users would appear at test stage with the ratings consisting of only ones and no zeros. We propose a unified graph signal sampling framework which enjoys the benefits of graph signal analysis and processing. The key idea is to transform each user's ratings on the items to a function (graph signal) on the vertices of an item-item graph, then learn structural graph properties to recover the function from its values on certain vertices — the problem of graph signal sampling. We propose a class of regularization functionals that takes into account discrete random label noise in the graph vertex domain, then develop the GS-IMC approach which biases the reconstruction towards functions that vary little between adjacent vertices for noise reduction. Theoretical result shows that accurate reconstructions can be achieved under mild conditions. For the online setting, we develop a Bayesian extension, i.e., BGS-IMC which considers continuous random Gaussian noise in the graph Fourier domain and builds upon a prediction-correction update algorithm to obtain the unbiased and minimum-variance reconstruction. Both GS-IMC and BGS-IMC have closed-form solutions and thus are highly scalable in large data as verified on public benchmarks.

## 1 INTRODUCTION

In domains such as recommender systems and social networks, only "likes" (i.e., ones) are observed in the system and service providers (e.g, Netflix) are interested in discovering potential "likes" for *existing users* to stimulate demand. This motivates the problem of 1-bit matrix completion (OBMC), of which the goal is to recover missing values in an $n$-by-$m$ item-user matrix $\mathbf{R} \in \{0, 1\}^{n \times m}$. We note that $\mathbf{R}_{i,j} = 1$ means that item $i$ is rated by user $j$, but $\mathbf{R}_{i,j} = 0$ is essentially unlabeled or unknown which is a mixture of unobserved positive examples and true negative examples.

However, in real world *new users*, who are not exposed to the model during training, may appear at testing stage. This fact stimulates the development of inductive 1-bit matrix completion, which aims to recover unseen vector $\mathbf{y} \in \{0, 1\}^n$ from its partial positive entries $\Omega_+ \subseteq \{j|\mathbf{y}_j = 1\}$ at test time. Fig. 1(a) emphasizes the difference between conventional and inductive approaches. More formally, let $\mathbf{M} \in \{0, 1\}^{n \times (m+1)}$ denote the underlying matrix, where only a subset of positive examples $\Psi$ is randomly sampled from $\{(i, j)|\mathbf{M}_{i,j} = 1, i \leq n, j \leq m\}$ such that $\mathbf{R}_{i,j} = 1$ for $(i, j) \in \Psi$ and $\mathbf{R}_{i,j} = 0$ otherwise. Consider $(m+1)$-th column $\mathbf{y}$ out of matrix $\mathbf{R}$, we likewise denote its observations $\mathbf{s}_i = 1$ for $i \in \Omega_+$ and $\mathbf{s}_i = 0$ otherwise. We note that the sampling process here assumes that there exists a random label noise $\boldsymbol{\xi}$ which flips a 1 to 0 with probability $\rho$, or equivalently $\mathbf{s} = \mathbf{y} + \boldsymbol{\xi}$ where

$$\boldsymbol{\xi}_i = -1 \text{ for } i \in \{j|\mathbf{y}_j = 1\} - \Omega_+, \quad \text{and } \boldsymbol{\xi}_i = 0 \text{ otherwise.} \tag{1}$$

Fig. 1(a) presents an example of $\mathbf{s}, \mathbf{y}, \boldsymbol{\xi}$ to better understand their relationships.

Fundamentally, the reconstruction of true $\mathbf{y}$ from corrupted $\mathbf{s}$ bears a resemblance with graph signal sampling. Fig. 1(b) shows that the item-user rating matrix $\mathbf{R}$ can be used to define a homogeneous

---

*Junchi Yan is the correspondence author who is also with Shanghai AI Laboratory. The work was in part supported by NSFC (62222607), STCSM (22511105100).

Figure 1: (a) Conventional 1-bit matrix completion focuses on recovering missing values in matrix $\mathbf{R}$, while inductive approaches aim to recover new column $\mathbf{y}$ from observations $\mathbf{s}$ that are observed at testing stage. $\boldsymbol{\xi}$ denotes discrete noise that randomly flips ones to zeros. (b) Our GS-IMC approach, which regards $\mathbf{y}$ as a signal residing on nodes of a *homogeneous item-item* graph, aims to reconstruct true signal $\mathbf{y}$ from its observed values (orange colored) on a subset of nodes (gray shadowed).

item-item graph (see Sec 3.1), such that user ratings $\mathbf{y}/\mathbf{s}$ on items can be regarded as signals residing on graph nodes. The reconstruction of bandlimited graph signals from certain subsets of vertices (see Sec 2) has been extensively studied in graph signal sampling (Pesenson, 2000; 2008).

Despite popularity in areas such as image processing (Shuman et al., 2013; Pang & Cheung, 2017; Cheung et al., 2018) and matrix completion (Romero et al., 2016; Mao et al., 2018; McNeil et al., 2021), graph signal sampling appears less studied in the specific *inductive one bit matrix completion* problem focused in this paper (see **Appendix A** for detailed related works). Probably most closely related to our approach are MRFCF (Steck, 2019) and SGMC (Chen et al., 2021) which formulate their solutions as *spectral graph filters*. However, we argue that these methods are orthogonal to us since they focus on optimizing the rank minimization problem, whereas we optimize the functional minimization problem, thereby making it more convinient and straightforward to process and analyze the matrix data with vertex-frequency analysis (Hammond et al., 2011; Shuman et al., 2013), time-variant analysis (Mao et al., 2018; McNeil et al., 2021), smoothing and filtering (Kalman, 1960; Khan & Moura, 2008). Furthermore, (Steck, 2019; Chen et al., 2021) can be incorporated as special cases of our unified graph signal sampling framework (see **Appendix B** for detailed discussions).

Another emerging line of research has focused on learning the mapping from side information (or content features) to latent factors (Jain & Dhillon, 2013; Xu et al., 2013; Ying et al., 2018; Zhong et al., 2019). However, it has been recently shown (Zhang & Chen, 2020; Ledent et al., 2021; Wu et al., 2021) that in general this family of algorithms would possibly suffer inferior expressiveness when high-quality content is not available. Further, collecting personal data is likely to be unlawful as well as a breach of the data minimization principle in GDPR (Voigt & Von dem Bussche, 2017).

Much effort has also been made to leverage the advanced graph neural networks (GNN) for improvements. van den Berg et al. (2017) represent the data matrix $\mathbf{R}$ by a bipartite graph then generalize the representations to unseen nodes by summing the embeddings over the neighbors. Zhang & Chen (2020) develop graph neural networks which encode the subgraphs around an edge into latent factors then decode the factors back to the value on the edge. Besides, Wu et al. (2021) consider the problem in a downsampled homogeneous graph (i.e., user-user graph in recommender systems) then exploit attention networks to yield inductive representations. The key advantage of our approach is not only the closed form solution which takes a small fraction of training time required for GNNs, but also theory results that guarantee accurate reconstruction and provide guidance for practical applications.

We emphasize the challenges when connecting ideas and methods of graph signal sampling with inductive 1-bit matrix completion — 1-bit quantization and online learning. Specifically, 1-bit quantization raises challenges for formulating the underlying optimization problems: minimizing squared loss on the observed positive examples $\Omega_+$ yields a degenerate solution — the vector with all entries equal to one achieves zero loss; minimizing squared loss on the corrupted data $\mathbf{s}$ introduces the systematic error due to the random label noise $\boldsymbol{\xi}$ in Eq. (1). To address the issue, we represent the observed data $\mathbf{R}$ as a homogeneous graph, then devise a broader class of regularization functionals on graphs to mitigate the impact of *discrete* random noise $\boldsymbol{\xi}$. Existing theory for total variation denoising (Sadhanala et al., 2016; 2017) and graph regularization (Belkin et al., 2004; Huang et al., 2011), which takes into account *continuous* Gaussian noise, does not sufficiently address recoverability in inductive 1-bit matrix completion (see Sec 3.4). We finally mange to derive a closed-form solution, entitled **G**raph **S**ampling for **I**nductive (1-bit) **M**atrix **C**ompletion GS-IMC which biases the reconstruction towards functions that vary little between adjacent vertices for noise reduction.

For online learning, existing matrix factorization methods (Devooght et al., 2015; Volkovs & Yu, 2015; He et al., 2016) incrementally update model parameters via gradient descent, requiring an expensive line search to set the best learning rate. To scale up to large data, we develop a Bayesian extension called BGS-IMC where a prediction-correction algorithm is devised to instantly refreshes the prediction given new incoming data. The prediction step tracks the evolution of the optimization problem such that the predicted iterate does not drift away from the optimum, while the correction step adjusts for the distance between current prediction and the new information at each step. The advantage over baselines is that BGS-IMC considers the uncertainties in the graph Fourier domain, and the prediction-correction algorithm can efficiently provide the unbiased and minimum-variance predictions in closed form, without using gradient descent techniques. The contributions are:

• **New Inductive 1-bit Matrix Completion Framework.** We propose and technically manage (for the first time to our best knowledge) to introduce graph signal sampling to inductive 1-bit matrix completion. It opens the possibility of benefiting the analysis and processing of the matrix with signal processing toolbox including vertex-frequency analysis (Hammond et al., 2011; Shuman et al., 2013), time-variant analysis (Mao et al., 2018; McNeil et al., 2021), smoothing and filtering (Kalman, 1960; Khan & Moura, 2008) etc. We believe that our unified framework can serve as a new paradigm for 1-bit matrix completion, especially in large-scale and dynamic systems.

• **Generalized Closed-form Solution.** We derive a novel closed-form solution (i.e., GS-IMC) in the graph signal sampling framework, which incorporates existing closed-form solutions as special cases, e.g., (Chen et al., 2021; Steck, 2019). GS-IMC is learned from only positive data with discrete random noise. This is one of key differences to typical denoising methods (Sadhanala et al., 2016) where efforts are spent on removing continuous Gaussian noise from a real-valued signal.

• **Robustness Enhancement.** We consider the online learning scenario and construct a Bayesian extension, i.e., BGS-IMC where a new prediction-correction algorithm is proposed to instantly yield unbiased and minimum-variance predictions given new incoming data. Experiments in **Appendix E** show that BGS-IMC is more cost-effective than many neural models such as SASREC (Kang & McAuley, 2018), BERT4REC (Sun et al., 2019) and GREC (Yuan et al., 2020). We believe that this proves a potential for the future application of graph signal sampling to sequential recommendation.

• **Theoretical Guarantee and Empirical Effectiveness.** We extend Paley-Wiener theorem of (Pesenson, 2009) on real-valued data to positive-unlabelled data with statistical noise. The theory shows that under mild conditions, unseen rows and columns in training can be recovered from a certain subset of their values that is present at test time. Empirical results on real-world data show that our methods achieve state-of-the-art performance for the challenging inductive Top-$N$ ranking tasks.

## 2 PRELIMINARIES

In this section, we introduce the notions and provide the necessary background of graph sampling theory. Let $\mathcal{G} = (V, E, w)$ denote a weighted, undirected and connected graph, where $V$ is a set of vertices with $|V| = n$, $E$ is a set of edges formed by the pairs of vertices and the positive weight $w(u, v)$ on each edge is a function of the similarity between vertices $u$ and $v$.

Space $L_2(\mathcal{G})$ is the Hilbert space of all real-valued functions $\mathbf{f} : V \rightarrow \mathbb{R}$ with the following norm:

$$\| \mathbf{f} \| = \sqrt{\sum_{v \in V} |\mathbf{f}(v)|^2},$$
(2)

and the discrete Laplace operator Ł is defined by the formula (Chung & Graham, 1997):

$$\mathbf{Ł f}(v) = \frac{1}{\sqrt{d(v)}} \sum_{u \in \mathcal{N}(v)} w(u, v) \left( \frac{\mathbf{f}(v)}{\sqrt{d(v)}} - \frac{\mathbf{f}(u)}{\sqrt{d(u)}} \right), \quad \mathbf{f} \in L_2(\mathcal{G})$$

where $\mathcal{N}(v)$ signifies the neighborhood of node $v$ and $d(v) = \sum_{u \in \mathcal{N}(v)} w(u, v)$ is the degree of $v$.

**Definition 1 (Graph Fourier Transform).** Given a function or signal $\mathbf{f}$ in $L_2(\mathcal{G})$, the graph Fourier transform and its inverse (Shuman et al., 2013) can be defined as follows:

$$\widetilde{\mathbf{f}}_{\mathcal{G}} = \mathbf{U}^\top \mathbf{f} \quad \text{and} \quad \mathbf{f} = \mathbf{U} \widetilde{\mathbf{f}},$$
(3)

Table 1: Regularization functions, operators, kernels with free parameters $\gamma \geq 0$, $a \geq 2$.

| | Function | Operator | Filter Kernel |
|---|---|---|---|
| **Tikhonov Regularization** (Tikhonov, 1963) | $R(\lambda) = \gamma\lambda$ | $R(Ł) = \gamma Ł$ | $H(\lambda) = 1/(1 + \gamma\lambda)$ |
| **Diffusion Process** (Stroock & Varadhan, 1969) | $R(\lambda) = \exp(\gamma/2\lambda)$ | $R(Ł) = \exp(\gamma/2Ł)$ | $H(\lambda) = 1/(\exp(\gamma/2\lambda) + 1)$ |
| **One-Step Random Walk** (Pearson, 1905) | $R(\lambda) = (a - \lambda)^{-1}$ | $R(Ł) = (aI - Ł)^{-}$ | $H(\lambda) = (a - \lambda)/(a - \lambda + 1)$ |
| **Inverse Cosine** (MacLane, 1947) | $R(\lambda) = (\cos \lambda\pi/4)^{-1}$ | $R(Ł) = (\cos Ł\pi/4)^{-}$ | $H(\lambda) = 1/(1/(\cos \lambda\pi/4) + 1)$ |

where $\mathbf{U}$ represents eigenfunctions of discrete Laplace operator $Ł$, $\widetilde{\mathbf{f}_\mathcal{G}}$ denotes the signal in the graph Fourier domain and $\widetilde{\mathbf{f}_\mathcal{G}}(\lambda_l) = \langle \mathbf{f}, \mathbf{u}_l \rangle$ signifies the information at the frequency $\lambda_l$[1].

**Definition 2 (Bandlimiteness).** $\mathbf{f} \in L_2(\mathcal{G})$ is called $\omega$-bandlimited function if its Fourier transform $\widetilde{\mathbf{f}_\mathcal{G}}$ has support in $[0, \omega]$, and $\omega$-bandlimited functions form the Paley-Wiener space $\text{PW}_\omega(\mathcal{G})$.

**Definition 3 (Graph Signal Sampling).** Given $\mathbf{y} \in \text{PW}_\omega(\mathcal{G})$, $\mathbf{y}$ can be recovered from its values on the vertices $\Omega_+$ by minimizing below objective (Pesenson, 2000; 2008), with positive scalar $k$:

$$\min_{\mathbf{f} \in L_2(\mathcal{G})} \parallel Ł^k \mathbf{f} \parallel \quad \text{s.t.,} \quad \mathbf{f}(v) = \mathbf{y}(v), \quad \forall v \in \Omega_+. \tag{4}$$

Recall that the observation in inductive 1-bit matrix completion consists of only ones and no zeros (i.e., $\mathbf{y}(v) = 1$ for $v \in \Omega_+$) and $\parallel Ł^k \mathbf{1} \parallel = 0$. It is obvious that minimizing the loss on the observed entries corresponding to ones, produces a degenerate solution — the vector with all entries equal to one achieves zero loss. From this point of view, existing theory for sampling real-valued signals (Pesenson, 2000; 2008) is not well suited to the inductive 1-bit matrix completion problem.

## 3 CLOSED-FORM SOLUTION FOR 1-BIT MATRIX COMPLETION

This section builds a unified graph signal sampling framework for inductive 1-bit matrix completion that can inductively recover $\mathbf{y}$ from positive ones on set $\Omega_+$. The rational behind our framework is that the rows that have similar observations are likely to have similar reconstructions. This makes a lot of sense in practice, for example a user (column) is likely to give similar items (rows) similar scores in recommender systems. To achieve this, we need to construct a homogeneous graph $\mathcal{G}$ where the connected vertices represent the rows which have similar observations, so that we can design a class of graph regularized functionals that encourage adjacent vertices on graph $\mathcal{G}$ to have similar reconstructed values. In particular, we mange to provide a closed-form solution to the matrix completion problem (entitled GS-IMC), together with theoretical bounds and insights.

### 3.1 GRAPH DEFINITION

We begin with the introduction of two different kinds of methods to construct *homogeneous* graphs by using the zero-one matrix $\mathbf{R} \in \mathbb{R}^{n \times m}$: (i) following the definition of hypergraphs (Zhou et al., 2007), matrix $\mathbf{R}$ can be regarded as the incidence matrix, so as to formulate the hypergraph Laplacian matrix as $Ł = \mathbf{I} - \mathbf{D}_v^{-1/2} \mathbf{R} \mathbf{D}_e^- \mathbf{R}^\top \mathbf{D}_v^{-1/2}$ where $\mathbf{D}_v \in \mathbb{R}^{n \times n}$ ($\mathbf{D}_e \in \mathbb{R}^{m \times m}$) is the diagonal degree matrix of vertices (edges); and (ii) for regular graphs, one of the most popular approaches is to utilize the covariance between rows to form the adjacent matrix $\mathbf{A}_{i,j} = \text{Cov}(\mathbf{R}_i, \mathbf{R}_j)$ for $i \neq j$ so that we can define the graph Laplacian matrix as $Ł = \mathbf{I} - \mathbf{D}_v^{-1/2} \mathbf{A} \mathbf{D}_v^{-1/2}$.

### 3.2 GRAPH SIGNAL SAMPLING FRAMEWORK

Given a graph $\mathcal{G} = (V, E)$, any real-valued column $\mathbf{y} \in \mathbb{R}^n$ can be viewed as a function on $\mathcal{G}$ that maps from $V$ to $\mathbb{R}$, and specifically the $i$-th vector component $\mathbf{y}_i$ is equivalent to the function value $\mathbf{y}(i)$ at the $i$-th vertex. Now it is obvious that the problem of inductive matrix completion, of which the goal is to recover column $\mathbf{y}$ from its values on entries $\Omega_+$, bears a resemblance to the problem of graph signal sampling that aims to recover function $\mathbf{y}$ from its values on vertices $\Omega_+$.

However, most of existing graph signal sampling methods (Romero et al., 2016; Mao et al., 2018; McNeil et al., 2021) yield degenerated solutions when applying them to the 1-bit matrix completion problem. A popular heuristic is to treat some or all of zeros as negative examples $\Omega_-$, then to recover $\mathbf{y}$ by optimizing the following functional minimization problem, given any $k = 2^l, l \in \mathbb{N}$:

$$\min_{\mathbf{f} \in L_2(\mathcal{G})} \parallel [R(Ł)]^k \mathbf{f} \parallel \quad \text{s.t.,} \quad \parallel \mathbf{s}_\Omega - \mathbf{f}_\Omega \parallel \leq \epsilon \tag{5}$$

---

[1]To be consistent with (Shuman et al., 2013), $\mathbf{u}_l$ ($l$-th column of matrix $\mathbf{U}$) is the $l$-th eigenvector associated with the eigenvalue $\lambda_l$, and the graph Laplacian eigenvalues carry a notion of frequency.

Figure 2: Recall results on Netflix data of very-high degree vertices (left), high degree vertices (left middle), medium degree vertices (right middle) and low degree vertices (right) for top-100 ranking tasks, where $\lambda_{50}$ on the x-axis corresponds to the assumption of space $\mathrm{PW}_{\lambda_{50}}(\mathcal{G})$ or namely we use the eigenfunctions whose eigenvalues are not greater than $\lambda_{50}$ to make predictions. The results show that *low(high)-frequency functions reflect user preferences on the popular (cold) items.*

where recall that $\mathbf{s} = \mathbf{y} + \boldsymbol{\xi}$ is the observed data corrupted by discrete random noise $\boldsymbol{\xi}$, and $\mathbf{s}_\Omega$ ($\mathbf{f}_\Omega$) signifies the values of $\mathbf{s}$ ($\mathbf{f}$) only on $\Omega = \Omega_+ \cup \Omega_-$; $R(\text{Ł}) = \sum_l R(\lambda_l)\mathbf{u}_l\mathbf{u}_l^\top$ denotes the regularized Laplace operator in which $\{\lambda_l\}$ and $\{\mathbf{u}_l\}$ are respectively the eigenvalues and eigenfunctions of operator Ł. It is worth noting that $\mathbf{s}(i) = \mathbf{y}(i) + \boldsymbol{\xi}(i) = 0$ for $i \in \Omega_-$ is not the true negative data, and hence $\Omega_-$ will introduce the systematic bias when there exists $i \in \Omega_-$ so that $\mathbf{y}(i) = 1$.

The choice of regularization function $R(\lambda)$ needs to account for two critical criteria: 1) The resulting regularization operator $R(\text{Ł})$ needs to be semi-positive definite. 2) As mentioned before, we expect the reconstruction $\hat{\mathbf{y}}$ to have similar values on adjacent nodes, so that the uneven functions should be penalized more than even functions. To account for this, we adopt the family of positive, monotonically increasing functions (Smola & Kondor, 2003) as present in Table 1.

To the end, we summarize two natural questions concerning our framework: 1) What are the benefits from introducing the regularized Laplacian penalty? It is obvious that minimizing the discrepancy between $\mathbf{s}_\Omega$ and $\mathbf{f}_\Omega$ does not provide the generalization ability to recover unknown values on the rest vertices $V - \Omega$, and Theorem 4 and 5 answer the question by examining the error bounds. 2) What kind of $R(\text{Ł})$ constitutes a reasonable choice? It has been studied in (Huang et al., 2011) that $R(\text{Ł})$ is most appropriate if it is unbiased, and an unbiased $R(\text{Ł})$ reduces variance without incurring any bias on the estimator. We also highlight the empirical study in **Appendix C** that evaluates how the performance is affected by the definition of graph $\mathcal{G}$ and regularization function $R(\lambda)$.

### 3.3 CLOSED-FORM SOLUTION

In what follows, we aim to provide a closed-form solution for our unified framework by treating all of the zeros as negative examples, i.e., $\mathbf{s}(v) = 1$ for $v \in \Omega_+$ and $\mathbf{s}(v) = 0$ otherwise. Then by using the method of Lagrange multipliers, we reformulate Eq. (5) to the following problem:

$$\min_{\mathbf{f} \in L_2(\mathcal{G})} \frac{1}{2} \langle \mathbf{f}, R(\text{Ł})\mathbf{f} \rangle + \frac{\varphi}{2} \|\mathbf{s} - \mathbf{f}\|^2, \tag{6}$$

where $\varphi > 0$ is a hyperparameter. Obviously, this problem has a **closed-form** solution:

$$\hat{\mathbf{y}} = \left(\mathbf{I} + R(\text{Ł})/\varphi\right)^- \mathbf{s} = \left(\sum_l \left(1 + R(\lambda_l)/\varphi\right)\mathbf{u}_l\mathbf{u}_l^\top\right)^- \mathbf{s} = H(\text{Ł})\mathbf{s}, \tag{7}$$

where $H(\text{Ł}) = \sum_l H(\lambda_l)\mathbf{u}_l\mathbf{u}_l^\top$ with kernel $1/H(\lambda_l) = 1 + R(\lambda)/\varphi$, and we exemplify $H(\lambda)$ when $\varphi = 1$ in Table 1. From the viewpoint of spectral graph theory, our GS-IMC approach is essentially a spectral graph filter that amplifies(attenuates) the contributions of low(high)-frequency functions.

**Remark.** To understand low-frequency and high-frequency functions, Figure 2 presents case studies in the context of recommender systems on the Netflix prize data (Bennett et al., 2007). Specifically, we divide the vertices (items) into four classes: very-high degree ($> 5000$), high degree ($> 2000$), medium degree ($> 100$) and low degree vertices. Then, we report the recall results of all the four classes in different Paley-Wiener spaces $\mathrm{PW}_{\lambda_{50}}(\mathcal{G}), \ldots, \mathrm{PW}_{\lambda_{1000}}(\mathcal{G})$ for top-100 ranking prediction. The interesting observation is: (1) the low-frequency functions with eigenvalues less than $\lambda_{100}$ contribute nothing to low degree vertices; and (2) the high-frequency functions whose eigenvalues are greater than $\lambda_{500}$ do not help to increase the performance on very-high degree vertices. This finding implies that low(high)-frequency functions reflect the user preferences on the popular(cold) items. From this viewpoint, the model defined in Eq. (7) aims to exploit the items with high click-through rate with high certainty, which makes sense in commercial applications.

### 3.4 ERROR ANALYSIS

Our GS-IMC approach defined in Eq. (7) bears a similarity to total variation denoising (Sadhanala et al., 2016; 2017), graph-constrained regularization (Belkin et al., 2004; 2006), and particularly Laplacian shrinkage methods (Huang et al., 2011). However, we argue that the proposed GS-IMC approach is fundamentally different from previous works. Specifically, they operate on real-valued data while GS-IMC deals with positive-unlabeled data. We believe that our problem setting is more complicated, since the unlabeled data is a mixture of unobserved positive examples and true negative examples. In addition, existing methods analyze the recoverability considering statistical noise to be *continuous* Gaussian, e.g., Theorem 3 (Sadhanala et al., 2016), Theorem 1.1 (Pesenson, 2009) etc.

However, we study upper bound of GS-IMC in the presence of *discrete* random label noise $\boldsymbol{\xi}$. Specifically, Theorem 4 extends Paley-Wiener theorem of (Pesenson, 2009) on real-valued data to positive-unlabelled data, showing that a bandlimited function $\mathbf{y}$ can be recovered from its values on certain set $\Omega$. Theorem 5 takes into account statistical noise $\boldsymbol{\xi}$ and shows that a bandlimited function $\mathbf{y}$ can be accurately reconstructed if $C_n^2 = C > 0$ is a constant, not growing with $n$.

**Theorem 4** (**Error Analysis, extension of Theorem 1.1 in (Pesenson, 2009)**). *Given $R(\lambda)$ with $\lambda \leq R(\lambda)$ on graph $\mathcal{G} = (V, E)$, assume that $\Omega^c = V - \Omega$ admits the Poincare inequality $\| \phi \| \leq \Lambda \| Ł\phi \|$ for any $\phi \in L_2(\Omega^c)$ with $\Lambda > 0$, then for any $\mathbf{y} \in \mathrm{PW}_\omega(\mathcal{G})$ with $0 < \omega \leq R(\omega) < 1/\Lambda$,*

$$\| \mathbf{y} - \hat{\mathbf{y}}_k \| \leq 2\Big(\Lambda R(\omega)\Big)^k \| \mathbf{y} \| \quad \text{and} \quad \mathbf{y} = \lim_{k \to \infty} \hat{\mathbf{y}}_k \tag{8}$$

*where $k$ is a pre-specified hyperparameter and $\hat{\mathbf{y}}_k$ is the solution of Eq. (5) with $\epsilon = 0$.*

**Remark.** Theorem 4 indicates that a better estimate of $\mathbf{y}$ can be achieved by simply using a higher $k$, but there is a trade-off between accuracy of the estimate on one hand, and complexity and numerical stability on the other. We found by experiments that GS-IMC with $k = 1$ can achieve SOTA results for inductive top-N recommendation on benchmarks. We provide more discussions in **Appendix G**.

**Theorem 5** (**Error Analysis, with label noise**). *Suppose that $\boldsymbol{\xi}$ is the random noise with flip rate $\rho$, and positive $\lambda_1 \leq \cdots \leq \lambda_n$ are eigenvalues of Laplacian $Ł$, then for any function $\mathbf{y} \in \mathrm{PW}_\omega(\mathcal{G})$,*

$$\mathbb{E}\Big[\mathrm{MSE}(\mathbf{y}, \hat{\mathbf{y}})\Big] \leq \frac{C_n^2}{n}\Big(\frac{\rho}{R(\lambda_1)(1 + R(\lambda_1)/\varphi)^2} + \frac{1}{4\varphi}\Big), \tag{9}$$

*where $C_n^2 = R(\omega) \| \mathbf{y} \|^2$, $\varphi$ is the regularization parameter and $\hat{\mathbf{y}}$ is defined in Eq. (7).*

**Remark.** Theorem 5 shows that for a constant $C_n^2 = C > 0$ (not growing with $n$), the reconstruction error converges to zero as $n$ is large enough. Also, the reconstruction error decreases with $R(\omega)$ declining which means low-frequency functions can be recovered more easily than high-frequency functions. We provide more discussions on $\varphi, \rho$ in **Appendix H**.

## 4 BAYESIAN GS-IMC FOR ONLINE LEARNING

In general, an inductive learning approach such as GAT (Veličković et al., 2017) and SAGE (Hamilton et al., 2017), etc., can naturally cope with the online learning scenario where the prediction is refreshed given a newly observed example. Essentially, GS-IMC is an inductive learning approach that can update the prediction, more effective than previous matrix completion methods (Devooght et al., 2015; He et al., 2016). Let $\Delta\mathbf{s}$ denote newly coming data that might be one-hot as in Fig. 3(a), $\hat{\mathbf{y}}$ denotes original prediction based on data $\mathbf{s}$, then we can efficiently update $\hat{\mathbf{y}}$ to $\hat{\mathbf{y}}_{\mathrm{new}}$ as follows:

$$\hat{\mathbf{y}}_{\mathrm{new}} = H(Ł)(\mathbf{s} + \Delta\mathbf{s}) = \hat{\mathbf{y}} + H(Ł)\Delta\mathbf{s}. \tag{10}$$

However, we argue that GS-IMC ingests the new data in an unrealistic, suboptimal way. Specifically, it does not take into account the model uncertainties, assuming that the observed positive data is noise-free. This assumption limits model's fidelity and flexibility for real applications. In addition, it assigns a uniform weight to each sample, assuming that the innovation, i.e., the difference between the current a priori prediction and the current observation information, is equal for all samples.

### 4.1 PROBLEM FORMULATION

To model the uncertainties, we denote a measurement by $\mathbf{z} = \mathbf{U}\hat{\mathbf{y}}$ (Fourier basis $\mathbf{U}$) which represents prediction $\hat{\mathbf{y}}$ in the graph Fourier domain and we assume that $\mathbf{z}$ is determined by a stochastic process.

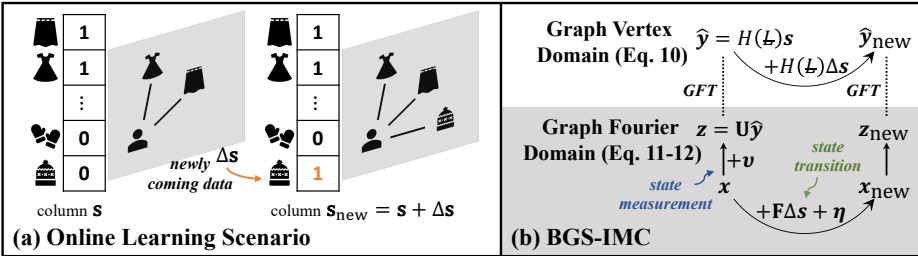

Figure 3: (a) Online learning scenario requires the model to refresh the predictions based on newly coming data $\Delta \mathbf{s}$ that is one-hot (orange colored). (b) GS-IMC deals with this problem in graph vertex domain using Eq. (10), while BGS-IMC operates in graph Fourier domain. The measurement $\mathbf{z}/\mathbf{z}_{\text{new}}$ is graph Fourier transformation of the prediction $\hat{\mathbf{y}}/\hat{\mathbf{y}}_{\text{new}}$, and we assume hidden states $\mathbf{x}/\mathbf{x}_{\text{new}}$ determine these measurements under noise $\nu$. To achieve this, $\mathbf{x}/\mathbf{x}_{\text{new}}$ should obey the evolution of $\hat{\mathbf{y}}/\hat{\mathbf{y}}_{\text{new}}$, and thus Eq. (11) represents Eq. (10) under noise $\eta$ in graph Fourier domain.

In Fig. 3(b), measurement $\mathbf{z}$ is governed by hidden state $\mathbf{x}$ and noise $\nu$ captures the data uncertainties in an implicit manner. The choice of state transition equation need to account for two critical criteria: (1) the model uncertainties need to be considered. (2) the transition from state $\mathbf{x}$ to state $\mathbf{x}_{\text{new}}$ need to represent the evolution of predictions $\hat{\mathbf{y}}/\hat{\mathbf{y}}_{\mathbf{y}}$ defined in Eq. (10).

To account for this, we propose a Bayesian extension of GS-IMC, entitled BGS-IMC, which considers the stochastic filtering problem in a dynamic state-space form:

$$\mathbf{x}_{\text{new}} = \mathbf{x} + \mathbf{F}\Delta \mathbf{s} + \eta \tag{11}$$

$$\mathbf{z}_{\text{new}} = \mathbf{x}_{\text{new}} + \nu \tag{12}$$

where Eq. (11) essentially follows Eq. (10) in the graph Fourier domain, i.e., multiplying both sides of Eq. (10) by $\mathbf{U}$. In control theory, $\mathbf{F} = \mathbf{U}H(Ł)$ is called the input matrix and $\Delta \mathbf{s}$ represents the system input vector. The state equation (11) describes how the true state $\mathbf{x}, \mathbf{x_{new}}$ evolves under the impact of the process noise $\eta \sim \mathcal{N}(0, \boldsymbol{\Sigma}_\eta)$, and the measurement equation (12) characterizes how a measurement $\mathbf{z}_{\text{new}} = \mathbf{U}^\top(\mathbf{s} + \Delta \mathbf{s})$ of the true state $\mathbf{x}_{\text{new}}$ is corrupted by the measurement noise $\nu \sim \mathcal{N}(0, \boldsymbol{\Sigma}_\nu)$. It is worth noting that larger determinant of $\boldsymbol{\Sigma}_\nu$ means that data points are more dispersed, while for $\boldsymbol{\Sigma}_\eta$ large determinant implies that BGS-IMC is not sufficiently expressive and it is better to use measurement for decision making, i.e., BGS-IMC is reduced to GS-IMC.

Using Bayes rule, the posterior is given by:

$$p(\mathbf{x}_{\text{new}}|\Delta \mathbf{s}, \mathbf{z}_{\text{new}}) \propto p(\mathbf{z}_{\text{new}}|\mathbf{x}_{\text{new}})p(\mathbf{x}_{\text{new}}|\Delta \mathbf{s}), \tag{13}$$

where $p(\mathbf{z}_{\text{new}}|\mathbf{x}_{\text{new}})$ and $p(\mathbf{x}_{\text{new}}|\Delta \mathbf{s})$ follow a Gauss-Markov process.

## 4.2 Prediction-Correction Update Algorithm

To make an accurate prediction, we propose a prediction-correction update algorithm, resembling workhorse Kalman filtering-based approaches (Kalman, 1960; Wiener et al., 1964). To our knowledge, the class of prediction-correction methods appears less studied in the domain of 1-bit matrix completion, despite its popularity in time-series forecasting (Simonetto et al., 2016; de Bézenac et al., 2020) and computer vision (Matthies et al., 1989; Scharstein & Szeliski, 2002).

In the prediction step, we follow the evolution of the state as defined in Eq. (11) to compute the mean and the covariance of conditional $p(\mathbf{x}_{\text{new}}|\Delta \mathbf{s})$:

$$\mathbb{E}[\mathbf{x}_{\text{new}}|\Delta \mathbf{s}] = \hat{\mathbf{x}} + \mathbf{F}\Delta \mathbf{s} = \bar{\mathbf{x}}_{\text{new}} \quad \text{and} \quad \text{Var}(\mathbf{x}_{\text{new}}|\Delta \mathbf{s}) = \mathbf{P} + \boldsymbol{\Sigma}_\eta = \bar{\mathbf{P}}_{\text{new}}, \tag{14}$$

where $\hat{\mathbf{x}}$ is the estimate state of $\mathbf{x}$ and $\mathbf{P}$ is the estimate covariance, i.e., $\mathbf{P} = \mathbb{E}(\mathbf{x} - \hat{\mathbf{x}})(\mathbf{x} - \hat{\mathbf{x}})^\top$, while $\bar{\mathbf{x}}_{\text{new}}, \bar{\mathbf{P}}_{\text{new}}$ are the extrapolated estimate state and covariance respectively. Meanwhile, it is easy to obtain the mean and the covariance of conditional $p(\mathbf{z}_{\text{new}}|\mathbf{x}_{\text{new}})$:

$$\mathbb{E}[\mathbf{z}_{\text{new}}|\mathbf{x}_{\text{new}}] = \mathbb{E}[\mathbf{x}_{\text{new}} + \nu] = \mathbf{x}_{\text{new}} \quad \text{and} \quad \text{Var}(\mathbf{z}_{\text{new}}|\mathbf{x}_{\text{new}}) = \mathbb{E}[\nu\nu^\top] = \boldsymbol{\Sigma}_\nu. \tag{15}$$

In the correction step, we combine Eq. (13) with Eq. (14) and (15):

$$p(\mathbf{x}_{\text{new}}|\Delta \mathbf{s}, \mathbf{z}_{\text{new}}) \propto \exp\left((\mathbf{x}_{\text{new}} - \mathbf{z}_{\text{new}})^\top \boldsymbol{\Sigma}_\nu^-(\mathbf{x}_{\text{new}} - \mathbf{z}_{\text{new}}) + (\mathbf{x}_{\text{new}} - \bar{\mathbf{x}}_{\text{new}})^\top \bar{\mathbf{P}}_{\text{new}}^-(\mathbf{x}_{\text{new}} - \bar{\mathbf{x}}_{\text{new}})\right).$$

By solving $\partial \ln p(\mathbf{x}_{\text{new}}|\Delta \mathbf{s}, \mathbf{z}_{\text{new}})/\partial \mathbf{x}_{\text{new}} = 0$, we have the following corrected estimate state $\hat{\mathbf{x}}_{\text{new}}$ and covariance $\mathbf{P}_{\text{new}}$, where we recall that the new measurement is defined as $\mathbf{z}_{\text{new}} = \mathbf{U}^{\top}(\mathbf{s} + \Delta \mathbf{s})$:

$$\hat{\mathbf{x}}_{\text{new}} = \bar{\mathbf{x}}_{\text{new}} + \mathbf{K}(\mathbf{z}_{\text{new}} - \bar{\mathbf{x}}_{\text{new}}) \tag{16}$$

$$\mathbf{P}_{\text{new}} = (\mathbf{I} - \mathbf{K})\bar{\mathbf{P}}_{\text{new}}(\mathbf{I} - \mathbf{K})^{\top} + \mathbf{K}\boldsymbol{\Sigma}_{\nu}\mathbf{K}^{\top} \tag{17}$$

$$\mathbf{K} = \bar{\mathbf{P}}_{\text{new}}(\bar{\mathbf{P}}_{\text{new}} + \boldsymbol{\Sigma}_{\nu})^{-}, \tag{18}$$

where $\mathbf{K}$ is the Kalman gain and $\mathbf{z}_{\text{new}} - \bar{\mathbf{x}}_{\text{new}}$ is called the innovation. It is worth noting that Eq. (16) adjusts the predicted iterate $\bar{\mathbf{x}}_{\text{new}}$ in terms of the innovation, the key difference to GS-IMC and existing methods, e.g., GAT (Veličković et al., 2017) and SAGE (Hamilton et al., 2017).

**Remark.** The BGS-IMC approach is highly scalable in Paley-Wiener spaces. Let $\text{PW}_{\omega}(\mathcal{G})$ be the span of $k$ ($\ll n$) eigenfunctions whose eigenvalues are no greater than $\omega$, then the transition matrix $\mathbf{F}$ in (11) is $k$-by-$n$ and every covariance matrix is of size $k \times k$. Computationally, when $\mathbf{P}, \boldsymbol{\Sigma}_{\eta}, \boldsymbol{\Sigma}_{\nu}$ are diagonal, it takes $\mathcal{O}(k^2)$ time to compute $\hat{\mathbf{x}}_{\text{new}}$ and $\mathbf{P}_{\text{new}}$, and $\mathcal{O}(nk)$ time for $\bar{\mathbf{x}}_{\text{new}}$ and $\bar{\mathbf{P}}_{\text{new}}$. The total time complexity is $\mathcal{O}(nk + k^2)$, linear to the number of vertices $n$. Further, Proposition 6 shows that $\hat{\mathbf{x}}_{\text{new}}$ in (16) is an unbiased and minimum-variance estimator.

**Proposition 6.** *Given an observation $\Delta \mathbf{s}$, provided $\mathbf{F}$ is known, $\hat{\mathbf{x}}_{\text{new}}$ obtained in Eq. (16) is the optimal linear estimator in the sense that it is unbiased and minimum-variance.*

To summarize, the complete procedure of BGS-IMC is to first specify $\boldsymbol{\Sigma}_{\eta}, \boldsymbol{\Sigma}_{\nu}, \mathbf{P}$ using prior knowledge, then to calculate extrapolated state $\bar{\mathbf{x}}_{\text{new}}$ using (14), and finally to obtain $\hat{\mathbf{x}}_{\text{new}}$ using (16) so that we have the updated model prediction as $\hat{\mathbf{y}}_{\text{new}} = \mathbf{U}\hat{\mathbf{x}}_{\text{new}}$ that ingests the new observation.

## 5 EXPERIMENT

This section evaluates GS-IMC (in Section 3) and BGS-IMC (in Section 4) on real-world datasets. All the experiments are conducted on the machines with Xeon 3175X CPU, 128G memory and P40 GPU with 24 GB memory. **The source code and models will be made publicly available**.

### 5.1 EXPERIMENTAL SETUP

We adopt three large real-world datasets widely used for evaluating recommendation algorithms: (1) Koubei ($1,828,250$ ratings of $212,831$ users and $10,213$ items); (2) Tmall ($7,632,826$ ratings of $320,497$ users and $21,876$ items); (3) Netflix ($100,444,166$ ratings of $400,498$ users and $17,770$ items). For each dataset, we follow the experimental protocols in (Liang et al., 2018; Wu et al., 2017a) for inductive top-N ranking, where the users are split into training/validation/test set with ratio $8:1:1$. Then, we use all the data from the training users to optimize the model parameters. In the testing phase, we sort all interactions of the validation/test users in chronological order, holding out the last one interaction for testing and inductively generating necessary representations using the rest data. The results in terms of hit-rate (HR) and normalized discounted cumulative gain (NDCG) are reported on the test set for the model which delivers the best results on the validation set.

We implement our method in Apache Spark with Intel MKL, where matrix computation is parallelized and distributed. In experiments, we denote item-user rating matrix by $\mathbf{R}$ and further define the Laplacian $Ł = \mathbf{I} - \mathbf{D}_v^{-1/2}\mathbf{R}\mathbf{D}_e^{-}\mathbf{R}^{\top}\mathbf{D}_v^{-1/2}$. We set $a=4$, $\gamma=1$, $\varphi=10$ for GS-IMC, while we set the covariance to $\boldsymbol{\Sigma}_{\eta}=\boldsymbol{\Sigma}_{\nu}=10^{-4}\mathbf{I}$ and initialize $\mathbf{P}$ using the validation data for BGS-IMC. In the test stage, if a user has $|\Omega|$ training interactions, BGS-IMC uses first $|\Omega|-1$ interactions to produce initial state $\hat{\mathbf{x}}$, then feed last interaction to simulate the online update.

In the literature, there are few of existing works that enable inductive inference for topN ranking only using the ratings. To make thorough comparisons, we prefer to strengthen IDCF with GCMC for the improved performance (IDCF+ for short) rather than report the results of IDCF (Wu et al., 2021) and GCMC (van den Berg et al., 2017) as individuals. Furthermore, we study their performance with different graph neural networks including ChebyNet (Defferrard et al., 2016), GAT (Veličković et al., 2017), GraphSage (Hamilton et al., 2017), SGC (Wu et al., 2019) and ARMA (Bianchi et al., 2021). We adopt the Adam optimizer (Kingma & Ba, 2015) with the learning rate decayed by $0.98$ every epoch. We search by grid the learning rate and $L_2$ regularizer in $\{0.1, 0.01, \dots, 0.00001\}$, the dropout rate over $\{0.1, 0.2, \dots, 0.7\}$ and the latent factor size ranging $\{32, 64, \dots, 512\}$ for the optimal performance. In addition, we also report the results of the shallow models i.e., MRCF (Steck, 2019) and SGMC (Chen et al., 2021) which are most closely related to our proposed method. The software provided by the authors is used in the experiments.

Table 2: **Hit-Rate** results against the baselines for inductive top-N ranking. Note that SGMC (Chen et al., 2021) is a special case of our method using the cut-off regularization, and MRFCF (Steck, 2019) is the full rank version of our method with (one-step) random walk regularization. The standard errors of the ranking metrics are **less than 0.005** for all the three datasets.

| Model | Koubei, Density=0.08% | | | Tmall, Density=0.10% | | | Netflix, Density=1.41% | | |
|---|---|---|---|---|---|---|---|---|---|
| | H@10 | H@50 | H@100 | H@10 | H@50 | H@100 | H@10 | H@50 | H@100 |
| **IDCF*** (Wu et al., 2021) | 0.14305 | 0.20335 | 0.24285 | 0.16100 | 0.27690 | 0.34573 | 0.08805 | 0.19788 | 0.29320 |
| **IDCF+GAT** (Veličković et al., 2017) | 0.19715 | 0.26440 | 0.30125 | **0.20033** | 0.32710 | 0.39037 | 0.08712 | 0.19387 | 0.27228 |
| **IDCF+GraphSAGE** (Hamilton et al., 2017) | 0.20600 | 0.27225 | 0.30540 | 0.19393 | 0.32733 | 0.39367 | 0.08580 | 0.19187 | 0.26972 |
| **IDCF+SGC** (Wu et al., 2019) | 0.20090 | 0.26230 | 0.30345 | 0.19213 | 0.32493 | 0.38927 | 0.08062 | 0.18080 | 0.26720 |
| **IDCF+ChebyNet** (Defferrard et al., 2016) | 0.20515 | 0.28100 | 0.32385 | 0.18163 | 0.32017 | 0.39417 | 0.08735 | 0.19335 | 0.27470 |
| **IDCF+ARMA** (Bianchi et al., 2021) | 0.20745 | 0.27750 | 0.31595 | 0.17833 | 0.31567 | 0.39140 | 0.08610 | 0.19128 | 0.27812 |
| **MRFCF** (Steck, 2019) | 0.17710 | 0.19300 | 0.19870 | 0.19123 | 0.28943 | 0.29260 | 0.08738 | 0.19488 | 0.29048 |
| **SGMC** (Chen et al., 2021) | 0.23290 | 0.31655 | 0.34500 | 0.13560 | 0.31070 | 0.40790 | 0.09740 | 0.22735 | 0.32193 |
| **GS-IMC** (ours, Sec 3) | 0.23460 | 0.31995 | 0.35065 | 0.13677 | 0.31027 | 0.40760 | 0.09725 | 0.22733 | 0.32225 |
| **BGS-IMC** (ours, Sec 4) | **0.24390** | **0.32545** | **0.35345** | 0.16733 | **0.34313** | **0.43690** | **0.09988** | **0.23390** | **0.33063** |

Table 3: **NDCG** results of GS-IMC and BGS-IMC against the baselines for inductive top-N ranking.

| Model | Koubei, Density=0.08% | | | Tmall, Density=0.10% | | | Netflix, Density=1.41% | | |
|---|---|---|---|---|---|---|---|---|---|
| | N@10 | N@50 | N@100 | N@10 | N@50 | N@100 | N@10 | N@50 | N@100 |
| **IDCF*** (Wu et al., 2021) | 0.13128 | 0.13992 | 0.14523 | 0.10220 | 0.12707 | 0.13821 | 0.05054 | 0.07402 | 0.08944 |
| **IDCF+GAT** (Veličković et al., 2017) | 0.15447 | 0.16938 | 0.17534 | **0.10564** | **0.13378** | 0.14393 | 0.04958 | 0.07250 | 0.08518 |
| **IDCF+GraphSAGE** (Hamilton et al., 2017) | 0.15787 | 0.17156 | 0.17701 | 0.10393 | 0.13352 | 0.14417 | 0.04904 | 0.07155 | 0.08419 |
| **IDCF+SGC** (Wu et al., 2019) | 0.15537 | 0.16848 | 0.17548 | 0.10287 | 0.13208 | 0.14260 | 0.04883 | 0.06965 | 0.08456 |
| **IDCF+ChebyNet** (Defferrard et al., 2016) | 0.15784 | 0.17406 | 0.18055 | 0.09916 | 0.12955 | 0.14175 | 0.04996 | 0.07268 | 0.08582 |
| **IDCF+ARMA** (Bianchi et al., 2021) | 0.15830 | 0.17320 | 0.17954 | 0.09731 | 0.12628 | 0.13829 | 0.04940 | 0.07192 | 0.08526 |
| **MRFCF** (Steck, 2019) | 0.10037 | 0.10410 | 0.10502 | 0.08867 | 0.11223 | 0.11275 | 0.05235 | 0.08047 | 0.09584 |
| **SGMC** (Chen et al., 2021) | 0.16418 | 0.18301 | 0.18764 | 0.07285 | 0.11110 | 0.12685 | 0.05402 | 0.08181 | 0.09710 |
| **GS-IMC** (ours, Sec 3) | 0.17057 | 0.18970 | 0.19468 | 0.07357 | 0.11115 | 0.12661 | 0.05504 | 0.08181 | 0.09759 |
| **BGS-IMC** (ours, Sec 4) | **0.17909** | **0.19680** | **0.20134** | 0.09222 | 0.13082 | **0.14551** | **0.05593** | **0.08400** | **0.09982** |

We omit the results of Markov chain Monte Carlo based FISM (He & McAuley, 2016), variational auto-encoder based MultVAE (Liang et al., 2018), scalable Collrank (Wu et al., 2017b), graph neural networks GCMC (van den Berg et al., 2017) and NGCF (Wang et al., 2019), as their accuracies were found below on par in SGMC (Chen et al., 2021) and IDCF (Wu et al., 2021).

### 5.2 Accuracy Comparison

In this section, GS-IMC and BGS-IMC assume that the underlying signal is $\lambda_{1000}$-bandlimited, and we compare them with eight state-of-the-arts graph based baselines, including spatial graph models (i.e., IDCF (Wu et al., 2021), IDCF+GAT (Veličković et al., 2017), IDCF+GraphSAGE (Hamilton et al., 2017)), approximate spectral graph models with high-order polynomials (i.e., IDCF+SGC (Wu et al., 2019), IDCF+ChebyNet (Defferrard et al., 2016), IDCF+ARMA (Bianchi et al., 2021)) and exact spectral graph models (i.e., MRFCF (Steck, 2019) and SGMC (Chen et al., 2021)).

In Table 2 and Table 3, the results on the real-world Koubei, Tmall and Netflix show that BGS-IMC outperforms all the baselines on all the datasets. Note that MRFCF (Steck, 2019) is the full rank version of GS-IMC with (one-step) random walk regularization. We can see that MRFCF underperforms its counterpart on all the three datasets, which demonstrates the advantage of the bandlimited assumption for inductive top-N ranking tasks. Further, BGS-IMC consistently outperforms GS-IMC on all three datasets by margin which proves the efficacy of the prediction-correction algorithm for incremental updates. Additionally, we provide extensive ablation studies in **Appendix C**, scalability studies in **Appendix D** and more comparisons with SOTA sequential models in **Appendix E**.

To summarize, the reason why the proposed method can further improve the prediction accuracy is due to **1)** GS-IMC exploits the structural information in the 1-bit matrix to mitigate the negative influence of *discrete* label noise in the graph *vertex* domain; and **2)** BGS-IMC further improves the prediction accuracy by considering *continuous* Gaussian noise in the graph *Fourier* domain and yielding unbiased and minimum-variance predictions using prediction-correction update algorithm.

## 6 Conclusion

We have introduced a unified graph signal sampling framework for inductive 1-bit matrix completion, together with theoretical bounds and insights. Specifically, GS-IMC is devised to learn the structural information in the 1-bit matrix to mitigate the negative influence of discrete label noise in the graph vertex domain. Second, BGS-IMC takes into account the model uncertainties in the graph Fourier domain and provides a prediction-correction update algorithm to obtain the unbiased and minimum-variance reconstructions. Both GS-IMC and BGS-IMC have closed-form solutions and are highly scalable. Experiments on the task of inductive top-N ranking have shown the supremacy.

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

## A    RELATED WORK

**Inductive matrix completion.** There has been a flurry of research on problem of inductive matrix completion (Chiang et al., 2018; Jain & Dhillon, 2013; Xu et al., 2013; Zhong et al., 2019), which leverage side information (or content features) in the form of feature vectors to predict inductively on new rows and columns. The intuition behind this family of algorithms is to learn mappings from the feature space to the latent factor space, such that inductive matrix completion methods can adapt to new rows and columns without retraining. However, it has been recently shown (Zhang & Chen, 2020; Ledent et al., 2021; Wu et al., 2021) that inductive matrix completion methods provide limited performance due to the inferior expressiveness of the feature space. On the other hand, the prediction accuracy has strong constraints on the content quality, but in practice the high quality content is becoming hard to collect due to legal risks (Voigt & Von dem Bussche, 2017). By contrast, one advantage of our approach is the capacity of inductive learning without using side information.

**Graph neural networks.** Inductive representation learning over graph structured data has received significant attention recently due to its ubiquitous applicability. Among the existing works, Graph-SAGE (Hamilton et al., 2017) and GAT (Veličković et al., 2017) propose to generate embeddings for previously unseen data by sampling and aggregating features from a node's local neighbors. In the meantime, various approaches such as ChebyNet (Defferrard et al., 2016) and GCN (Kipf & Welling, 2016) exploit convolutional neural networks to capture sophisticated feature information but are generally less scalable. To address the scalability issue, Wu et al. (2019) develop simplified graph convolutional networks (SGCN) which utilize polynomial filters to simulate the stacked graph convolutional layers. Furthermore, Bianchi et al. (2021) extend auto-regressive moving average (ARMA) filters to convolutional layers for broader frequency responses.

To leverage recent advance in graph neural networks, lightGCN (He et al., 2020), GCMC (van den Berg et al., 2017) and PinSAGE (Ying et al., 2018) represent the matrix by a bipartite graph then generalize the representations to unseen nodes by summing the content-based embeddings over the neighbors. Differently, IGMC (Zhang & Chen, 2020) trains graph neural networks which encode the subgraphs around an edge into latent factors then decode the factors back to the value on the edge. Recently, IDCF (Wu et al., 2021) studies the problem in a downsampled homogeneous graph (i.e., user-user graph in recommender systems) then applies attention networks to yield inductive representations. Probably most closely related to our approach are IDCF (Wu et al., 2021) and IGMC (Zhang & Chen, 2020) which do not assume any side information, such as user profiles and item properties. The key advantage of our approach is not only the closed form solution for efficient GNNs training, but also the theoretical results which guarantee the reconstruction of unseen rows and columns and the practical guidance for potential improvements.

**Graph signal sampling.** In general, graph signal sampling aims to reconstruct real-valued functions defined on the vertices (i.e., graph signals) from their values on certain subset of vertices. Existing approaches commonly build upon the assumption of *bandlimitedness*, by which the signal of interest lies in the span of leading eigenfunctions of the graph Laplacian (Pesenson, 2000; 2008). It is worth noting that we are not the first to consider the connections between graph signal sampling and matrix completion, as recent work by Romero et al. (Romero et al., 2016) has proposed a unifying kernel based framework to broaden both of graph signal sampling and matrix completion perspectives. However, we argue that Romero's work and its successors (Benzi et al., 2016; Mao et al., 2018; McNeil et al., 2021) are orthogonal to our approach as they mainly focus on real-valued matrix completion in the transductive manner. Specifically, our approach concerns two challenging problems when connecting the ideas and methods of graph signal sampling with inductive one-bit matrix completion — *one-bit quantization and online learning*.

To satisfy the requirement of online learning, existing works learn the parameters for new rows and columns by performing either stochastic gradient descent used in MCEX (Giménez-Febrer et al., 2019), or alternating least squares used in eALS (He et al., 2016). The advantage of BGS-IMC is three fold: (i) BGS-IMC has closed form solutions, bypassing the well-known difficulty for tuning

learning rate; and (ii) BGS-IMC considers the random Gaussian noise in the graph Fourier domain, characterizing the uncertainties in the measurement and modeling; (iii) prediction-correction algorithm, resembling Kalman filtering, can provide unbiased and minimum-variance reconstructions.

Probably most closely related to our approach are SGMC (Chen et al., 2021) and MRFCF (Steck, 2019) in the sense that both of them formulate their solutions as *spectral graph filters* and can be regarded as methods for data filtering in domains of discrete signal processing. More specifically, SGMC optimizes latent factors $\mathbf{V}, \mathbf{U}$ by minimizing the normalized matrix reconstruction error:

$$\min_{\mathbf{U},\mathbf{V}} \parallel \mathbf{D}_v^{-1/2}\mathbf{R}\mathbf{D}_e^{-1/2} - \mathbf{V}\mathbf{U} \parallel, \quad \text{s.t.} \quad \parallel \mathbf{U} \parallel \leq \epsilon, \parallel \mathbf{V} \parallel \leq \eta, \tag{19}$$

while MRFCF minimizes the following matrix reconstruction error:

$$\min_{\mathbf{X}} \parallel \mathbf{R} - \mathbf{X}\mathbf{R} \parallel + \lambda \parallel \mathbf{X} \parallel \quad \text{s.t.} \quad \text{diag}(\mathbf{X}) = 0, \tag{20}$$

where the diagonal entries of parameter $\mathbf{X}$ is forced to zero. It is obvious now that both SGMC and MRFCF focus on minimizing the matrix reconstruction problem. This is one of the key differences to our graph signal sampling framework which optimizes the functional minimization problem as defined in Eq. 5. We argue that our problem formulation is more suitable for the problem of inductive one-bit matrix completion, since it focuses on the reconstruction of bandlimited functions, no matter if the function is observed in the training or at test time. Perhaps more importantly, both of methods (Chen et al., 2021; Steck, 2019) can be included as special cases of our framework. *We believe that a unified framework cross graph signal sampling and inductive matrix completion could benefit both fields, since the modeling knowledge from both domains can be more deeply shared.*

**Advantages of graph signal sampling perspectives.** A graph signal sampling perspective requires to model 1-bit matrix data as signals on a graph and formulate the objective in the functional space. Doing so opens the possibility of processing, filtering and analyzing the matrix data with vertex-frequency analysis (Hammond et al., 2011; Shuman et al., 2013), time-variant analysis (Mao et al., 2018; McNeil et al., 2021), smoothing and filtering (Kalman, 1960; Khan & Moura, 2008) etc. In this paper, we technically explore the use of graph spectral filters to inductively recover the missing values of matrix, Kalman-filtering based approach to deal with the streaming data in online learning scenario, and vertex-frequency analysis to discover the advantages of dynamic BERT4REC model over static BGS-IMC model. We believe that our graph signal sampling framework can serve as a new paradigm for 1-bit matrix completion, especially in large-scale and dynamic systems.

## B   GENERALIZING SGMC AND MRFCF

This section shows how GS-IMC generalizes SGMC (Chen et al., 2021) and MRFCF (Steck, 2019).

**GS-IMC generalizes SGMC.** Given the observation $\mathbf{R}$, we follow standard routine of hypergraph (Zhou et al., 2007) to calculate the hypergraph Laplacian matrix $\mathbf{Ł} = \mathbf{I} - \mathbf{D}_v^{-1/2}\mathbf{R}\mathbf{D}_e^{-}\mathbf{R}^\top\mathbf{D}_v^{-1/2}$, where $\mathbf{D}_v$ ($\mathbf{D}_e$) is the diagonal degree matrix of vertices (edges). Then the rank-$k$ approximation (see Eq. (9) in (Chen et al., 2021)) is equivalent to our result using bandlimited norm $R(\lambda) = 1$ if $\lambda \leq \lambda_k$ and $R(\lambda) = \infty$ otherwise,

$$\hat{\mathbf{y}} = \Big( \sum_l \Big(1 + R(\lambda_l)/\varphi\Big)\mathbf{u}_l\mathbf{u}_l^\top \Big)^{-}\mathbf{s} = \sum_{l \leq k} \mathbf{u}_l\mathbf{u}_l^\top\mathbf{s} = \mathbf{U}_k\mathbf{U}_k^\top\mathbf{s}$$

where we set $\varphi = \infty$ and $\lim_{\varphi \to \infty} R(\lambda)/\varphi = \infty$ for $\lambda > \lambda_k$, and matrix $\mathbf{U}_k$ comprises $k$ leading eigenvectors whose eigenvalues are less than or equal to $\lambda_k$.

**GS-IMC generalizes MRFCF.** Given $\mathbf{R}$, we simply adopt the correlation relationship to construct the affinity matrix and define the Laplacian as $\mathbf{Ł} = 2\mathbf{I} - \mathbf{D}_v^{-1/2}\mathbf{R}\mathbf{R}^\top\mathbf{D}_v^{-1/2}$. Then the matrix approximation (see Eq. (4) in (Steck, 2019)) is equivalent to our GS-IMC approach using one-step

random walk norm,

$$\hat{\mathbf{y}} = \Big( \sum_l \Big( 1 + \frac{1}{a - \lambda} \Big) \mathbf{u}_l \mathbf{u}_l^\top \Big)^- \mathbf{s}$$

$$= \sum_l \Big( 1 - \frac{1}{a - \lambda + 1} \Big) \mathbf{u}_l \mathbf{u}_l^\top \mathbf{s}$$

$$= \Big\{ \mathbf{I} - \big( (a+1)\mathbf{I} - \textit{Ł} \big)^- \Big\} \mathbf{s}$$

$$= \Big\{ \mathbf{I} - \big( (a-1)\mathbf{I} + \mathbf{D}_v^{1/2} \mathbf{R}\mathbf{R}^\top \mathbf{D}_v^{1/2} \big)^- \Big\} \mathbf{s}$$

where we set $\varphi = 1$ and $a \geq \lambda_{\max}$ is a pre-specified parameter for the random walk regularization.

## C  ABLATION STUDIES

This study evaluates how GS-IMC and BGS-IMC perform with different choice of the regularization function and the graph definition. In the following, we assume the underlying signal to recover is in the Paley-Wiener space $\mathrm{PW}_{\lambda_{1000}}(\mathcal{G})$, and hence we only take the first 1000 eigenfunctions whose eigenvalues are not greater than $\lambda_{1000}$ to make predictions.

### C.1  IMPACT OF REGULARIZATION FUNCTIONS

Table 4 and 5 show that for the proposed GS-IMC models, Tikhonov regularization produces the best HR and NDCG results on both Koubei and Netflix, while Diffusion process regularization performs the best on Tmall. Meanwhile, BGS-IMC with random walk regularization achieves the best HR and NDCG results on Koubei, while Tikhonov regularization and Diffusion process regularization are best on Tmall and Netflix. Perhaps more importantly, BGS-IMC consistently outperforms GS-IMC on all three datasets by margin which proves the efficacy of the prediction-correction algorithm.

We highlight the reason why BGS-IMC can further improve the performance of GS-IMC is due to the fact that BGS-IMC considers Gaussian noise in the Fourier domain and the prediction-correction update algorithm is capable of providing unbiased and minimum-variance predictions.

Table 4: **Hit-Rate** of GS-IMC, BGS-IMC with different regularization $R(\lambda)$ for inductive top-N ranking. Overall, BGS-IMC consistently outperforms GS-IMC. The standard errors of the ranking metrics are **less than 0.005** for all the three datasets.

| Model | Koubei, Density=0.08% | | | Tmall, Density=0.10% | | | Netflix, Density=1.41% | | |
|---|---|---|---|---|---|---|---|---|---|
| | H@10 | H@50 | H@100 | H@10 | H@50 | H@100 | H@10 | H@50 | H@100 |
| **GS-IMC-Tikhonov** (Sec. 3) | 0.23430 | 0.31995 | 0.35065 | 0.13363 | 0.30630 | 0.40370 | 0.09725 | 0.22733 | 0.32190 |
| **GS-IMC-Diffusion Process** (Sec. 3) | 0.23460 | 0.31440 | 0.34370 | 0.13677 | 0.30863 | 0.40390 | 0.09678 | 0.21980 | 0.31750 |
| **GS-IMC-Random Walk** (Sec. 3) | 0.23360 | 0.31860 | 0.34935 | 0.13423 | 0.30853 | 0.40550 | 0.09660 | 0.22328 | 0.32235 |
| **GS-IMC-Inverse Cosine** (Sec. 3) | 0.23300 | 0.31710 | 0.34645 | 0.13537 | 0.31027 | 0.40760 | 0.09675 | 0.22575 | 0.32225 |
| **BGS-IMC-Tikhonov** (Sec. 4) | 0.24260 | 0.32320 | 0.35045 | **0.16733** | **0.34313** | **0.43690** | **0.09988** | **0.23390** | **0.33063** |
| **BGS-IMC-Diffusion Process** (Sec. 4) | 0.24385 | 0.32185 | 0.34910 | 0.16680 | 0.34263 | 0.43317 | 0.09853 | 0.22630 | 0.32450 |
| **BGS-IMC-Random Walk** (Sec. 4) | **0.24390** | **0.32545** | **0.35345** | 0.16303 | 0.34127 | 0.43447 | 0.09825 | 0.23028 | 0.32973 |
| **BGS-IMC-Inverse Cosine** (Sec. 4) | 0.24275 | 0.32405 | 0.35130 | 0.16567 | 0.34303 | 0.43637 | 0.09945 | 0.23260 | 0.33055 |

Table 5: **NDCG** of GS-IMC, BGS-IMC with different regularization for inductive top-N ranking.

| Model | Koubei, Density=0.08% | | | Tmall, Density=0.10% | | | Netflix, Density=1.41% | | |
|---|---|---|---|---|---|---|---|---|---|
| | N@10 | N@50 | N@100 | N@10 | N@50 | N@100 | N@10 | N@50 | N@100 |
| **GS-IMC-Tikhonov** (Sec. 3) | 0.17057 | 0.18970 | 0.19468 | 0.07174 | 0.10940 | 0.12519 | 0.05399 | 0.08181 | 0.09709 |
| **GS-IMC-Diffusion Process** (Sec. 3) | 0.16943 | 0.18742 | 0.19219 | 0.07357 | 0.11115 | 0.12661 | 0.05504 | 0.08134 | 0.09713 |
| **GS-IMC-Random Walk** (Sec. 3) | 0.16846 | 0.18753 | 0.19253 | 0.07208 | 0.11011 | 0.12582 | 0.05452 | 0.08158 | 0.09759 |
| **GS-IMC-Inverse Cosine** (Sec. 3) | 0.16560 | 0.18453 | 0.18930 | 0.07265 | 0.11083 | 0.12660 | 0.05410 | 0.08173 | 0.09734 |
| **BGS-IMC-Tikhonov** (Sec. 4) | 0.17540 | 0.19352 | 0.19794 | 0.09144 | 0.13021 | 0.14544 | 0.05535 | **0.08400** | **0.09982** |
| **BGS-IMC-Diffusion Process** (Sec. 4) | **0.17909** | 0.19664 | 0.20108 | **0.09222** | **0.13082** | **0.14551** | **0.05593** | 0.08321 | 0.09909 |
| **BGS-IMC-Random Walk** (Sec. 4) | 0.17854 | **0.19680** | **0.20134** | 0.08956 | 0.12873 | 0.14387 | 0.05533 | 0.08349 | 0.09958 |
| **BGS-IMC-Inverse Cosine** (Sec. 4) | 0.17625 | 0.19451 | 0.19894 | 0.09094 | 0.12992 | 0.14507 | 0.05546 | 0.08394 | 0.09964 |

Table 6: **HR**, **NDCG** on the Netflix prize data of GS-IMC (w/ random walk regularization), where we adopt different methods for constructing the homogeneous graph for inductive top-N ranking.

| | HR@10 | HR@50 | HR@100 | NDCG@10 | NDCG@50 | NDCG@100 |
|---|---|---|---|---|---|---|
| **GS-IMC w/ Hypergraph** | 0.09660±0.0006 | 0.22328±0.0002 | 0.32235±0.0011 | 0.05452±0.0004 | 0.08158±0.0004 | 0.09759±0.0002 |
| **GS-IMC w/ Covariance** | 0.09767±0.0012 | 0.22388±0.0006 | 0.31312±0.0052 | 0.05454±0.0005 | 0.08171±0.0007 | 0.09613±0.0007 |

## C.2 IMPACT OF GRAPH DEFINITIONS

Table 6 present the HR and NDCG results of GS-IMC with one-step random walk regularization on the Netflix prize data. To avoid the clutter, we omit the results of GS-IMC with other regularization functions, since their results share the same trends. It seems that the regular graph that use covariance matrix as the affinity matrix has better HR and NDCG results when recommending 10 and 50 items, while the hypergraph helps achieve better results when recommending 100 items.

## D SCALABILITY STUDIES

The solution for either GS-IMC or BGS-IMC requires to compute leading eigenvetors whose eigenvalues are less than or equal to pre-specified $\omega$. However, one might argue that it is computationally intractable on the industry-scale datasets. To address the concerns, one feasible approach is to perform the Nyström (Fowlkes et al., 2004) method to obtain the leading eigenvectors. For the completeness of the paper, we present the pseudo-code of the approximate eigendecomposition (Chen et al., 2021) in Algorithm 1, of which the computational complexity is $\mathcal{O}(lnk + k^3)$ where $n$ is the number of columns in Ł, $l$ is the number of sampled columns and $k$ is the number of eigenvectors to compute. This reduces the overhead from $\mathcal{O}(n^3)$ to $\mathcal{O}(lnk + k^3)$, linear to the number of rows.

To evaluate how the proposed GS-IMC and BGS-IMC methods perform with the approximate eigenvectors, we conduct the experiments on the largest Netflix prize data. Table 7 reports the HR, NDCG and runtime results for the standard GS-IMC and BGS-IMC methods, and their scalable versions entitled GS-IMCs and BGS-IMCs. To make the comparison complete, we also present the results of neural IDCF (Wu et al., 2021) model equipped with ChebyNet (Defferrard et al., 2016). It is obvious that the standard GS-IMC and BGS-IMC methods consume only a small fraction of training time, required by graph neural networks. Meanwhile, GS-IMCs achieves comparable ranking

---

**Algorithm 1** Approximate Eigendecomposition

**Require:** $n \times l$ matrix $\mathbf{C}$ derived from $l$ columns sampled from $n \times n$ kernel matrix $\mathbf{L}$ without replacement, $l \times l$ matrix $\mathbf{A}$ composed of the intersection of these $l$ columns, $l \times l$ matrix $\mathbf{W}$, rank $k$, the oversampling parameter $p$ and the number of power iterations $q$.

**Ensure:** approximate eigenvalues $\widetilde{\mathbf{\Sigma}}$ and eigenvectors $\widetilde{\mathbf{U}}$.

1: Generate a random Gaussian matrix $\mathbf{\Omega} \in \mathbb{R}^{l \times (k+p)}$, then compute the sample matrix $\mathbf{A}^q \mathbf{\Omega}$.
2: Perform QR-Decomposition on $\mathbf{A}^q \mathbf{\Omega}$ to obtain an orthonormal matrix $\mathbf{Q}$ that satisfies the equation $\mathbf{A}^q \mathbf{\Omega} = \mathbf{Q}\mathbf{Q}^\top \mathbf{A}^q \mathbf{\Omega}$, then solve $\mathbf{Z}\mathbf{Q}^\top \mathbf{\Omega} = \mathbf{Q}^\top \mathbf{W}\mathbf{\Omega}$.
3: Compute the eigenvalue decomposition on the $(k + p)$-by-$(k + p)$ matrix $\mathbf{Z}$, i.e., $\mathbf{Z} = \mathbf{U_Z}\mathbf{\Sigma_Z}\mathbf{U_Z}^\top$, to obtain $\mathbf{U}_W = \mathbf{Q}\mathbf{U}_Z[:, :k]$ and $\mathbf{\Sigma}_W = \mathbf{\Sigma}_Z[: k, : k]$.
4: Return $\widetilde{\mathbf{\Sigma}} \leftarrow \mathbf{\Sigma}_W, \widetilde{\mathbf{U}} \leftarrow \mathbf{CA}^{-1/2}\mathbf{U}_W\mathbf{\Sigma}_W^{-1/2}$.

---

Table 7: **Hit-Rate**, **NDCG** and **Runtime** of the enhanced IDCF (Wu et al., 2021) model equipped with ChebyNet (Defferrard et al., 2016), GS-IMC, BGS-IMC (w/ random walk regularization) and their scalable versions (i.e., GS-IMCs and BGS-IMCs) for inductive top-N ranking on Netflix data.

| | HR@10 | HR@50 | HR@100 | NDCG@10 | NDCG@50 | NDCG@100 | Runtime |
|---|---|---|---|---|---|---|---|
| **IDCF+ChebyNet** | 0.08735±0.0016 | 0.19335±0.0042 | 0.27470±0.0053 | 0.04996±0.0010 | 0.07268±0.0017 | 0.08582±0.0037 | 598 min |
| **GS-IMC** | 0.09660±0.0006 | 0.22328±0.0002 | 0.32235±0.0011 | 0.05452±0.0004 | 0.08158±0.0004 | 0.09759±0.0002 | 12.0 min |
| **GS-IMCs** | 0.09638±0.0007 | 0.22258±0.0009 | 0.31994±0.0015 | 0.05352±0.0006 | 0.08135±0.0006 | 0.09657±0.0002 | 1.5 min |
| **BGS-IMC** | 0.09988±0.0006 | 0.23390±0.0005 | 0.33063±0.0009 | 0.05593±0.0004 | 0.08400±0.0004 | 0.09982±0.0001 | 12.5 min |
| **BGS-IMCs** | 0.10005±0.0011 | 0.23318±0.0014 | 0.32750±0.0020 | 0.05508±0.0006 | 0.08365±0.0006 | 0.09890±0.0001 | 2.0 min |

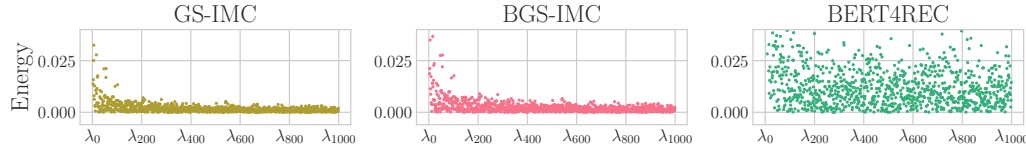

Figure 4: Spectrum analysis for static GS-IMC, BGS-IMC and dynamic BERT4REC on the Koubei dataset. Compared to BERT4REC, the energy of GS-IMC and BGS-IMC is concentrated on the low frequencies since the high-frequency functions are highly penalized during minimization.

performance to GS-IMC, while improving the efficiency by 8X. Likewise, BGS-IMCs enjoys the improvement in the system scalability without significant loss in prediction accuracy. The overall results demonstrate that GS-IMC and BGS-IMC are highly scalable in very large data.

## E    SPECTRUM ANALYSIS AND DISCUSSION WITH SEQUENTIAL MODELS

We compare BGS-IMC with recent sequential recommendation models, including Transformer-based SAS-REC (Kang & McAuley, 2018), BERT-based BERT4REC (Sun et al., 2019) and causal CNN based GREC (Yuan et al., 2020). We choose the embedding size of 256 and search the optimal hyper-parameters by grid. Each model is configured using the

Table 8: Comparisons to neural sequential models for the task of inductive top-N ranking on Koubei.

| Model | BERT4REC | SASREC | GREC | BGS-IMC |
|---|---|---|---|---|
| H@10 | 0.23760 | 0.23015 | 0.22405 | **0.24390** |
| H@50 | 0.32385 | 0.30500 | 0.30065 | **0.32545** |
| H@100 | **0.35965** | 0.34735 | 0.34350 | 0.35345 |
| N@10 | **0.18822** | 0.18496 | 0.18118 | 0.17854 |
| N@50 | **0.20663** | 0.20137 | 0.19816 | 0.19680 |
| N@100 | **0.21402** | 0.20819 | 0.20583 | 0.20134 |
| Runtime | 89 min | 37 min | 27 min | **83 sec** |

same parameters provided by the original paper i.e., two attention blocks with one head for SAS-REC, three attention blocks with eight heads for BERT4REC and six dilated CNNs with degrees $1, 2, 2, 4, 4, 8$ for GREC.

Table 8 presents HR and NDCG results on Koubei for inductive top-N ranking. Note that BGS-IMC only accepts the most recent behavior to update the obsolete state for incremental learning, whereas SASREC, BERT4REC and GREC focus on modeling the dynamic patterns in the sequence. Hence, such a comparison is not in favor of BGS-IMC. Interestingly, we see that static BGS-IMC achieves comparable HR results to SOTA sequential models, while consuming a small fraction of running time. From this viewpoint, BGS-IMC is more cost-effective than the compared methods.

To fully understand the performance gap in NDCG, we analyze GS-IMC, BGS-IMC and the best baseline BERT4REC in the graph spectral domain, where we limit the $\ell_2$ norm of each user's spectral signals to one and visualize their averaged values in Figure 4. As expected, the energy of GS-IMC and BGS-IMC is concentrated on the low frequencies, since the high-frequency functions are highly penalized during minimization. Furthermore, the proposed prediction-correction update algorithm increases the energy of high-frequency functions. This bears a similarity with BERT4REC of which high-frequency functions are not constrained and can aggressively raise the rankings of unpopular items. This explains why BERT4REC and BGS-IMC have better NDCGs than GS-IMC.

## F    LIMITATION AND FUTURE WORK

**Limitation on sequence modeling.** The proposed BGS-IMC method is simple and cannot capture the sophisticated dynamics in the sequence. However, we believe that our work opens the possibility of benefiting sequential recommendation with graph signal processing techniques, for example extended Kalman filter, KalmanNet and Particle filter.

**Limitation on sample complexity.** The sample complexity is not provided in the paper, and we believe that this is an open problem due to the lack of regularity in the graph which prevent us from defining the idea of sampling "every other node" (the reader is referred to (Anis et al., 2016; Ortega et al., 2018) for more details).

**Future work on deep graph learning.** Though GS-IMC and BGS-IMC are mainly compared with neural graph models, we note that our approach can help improve the performance of existing graph neural networks including GAT (Veličković et al., 2017) and SAGE (Hamilton et al., 2017), etc. We summarize the following directions for future works: **1)** It is interesting to see how GS-IMC takes advantage of content features. One feasible idea is to use GS-IMC as multi-scale wavelets which

can be easily adapted to graph neural networks; **2)** BGS-IMC can also be utilized to optimize the aggregation module for the improved robustness, as every neighbor's representation can be viewed as a measurement of the query node's representation.

## G    PROOF OF THEOREM 4

*Proof.* This proof is analogous to Theorem 1.1 in (Pesenson, 2009), where we extend their results from Sobolev norm to a broader class of positive, monotonically increasing functionals.

**Proof of the first part of the Theorem 4.**

Suppose that the Laplacian operator Ł has bounded inverse and the fitting error $\epsilon = 0$, if $\mathbf{y} \in \mathrm{PW}_\omega(\mathcal{G})$ and $\hat{\mathbf{y}}_k$ interpolate $\mathbf{y}$ on a set $\Omega = V - \Omega^c$ and $\Omega^c$ admits the Poincare inequality $\| \phi \| \leq \Lambda \| Ł\phi \|$ for any $\phi \in L_2(\Omega^c)$. Then $\mathbf{y} - \hat{\mathbf{y}}_k \in L_2(\Omega^c)$ and we have

$$\|\mathbf{y} - \hat{\mathbf{y}}_k\| \leq \Lambda \|Ł(\mathbf{y} - \hat{\mathbf{y}}_k)\|.$$

At this point, we can apply Lemma 7 with $\Lambda = a$ and $\phi = \mathbf{y} - \hat{\mathbf{y}}_k$. It gives the following inequality

$$\| \mathbf{y} - \hat{\mathbf{y}}_k \| \leq \Lambda^k \| Ł^k(\mathbf{y} - \hat{\mathbf{y}}_k) \|$$

for all $k = 2^l, l = 0, 1, 2, \ldots$ Since $R(\lambda)$ is positive and monotonically increasing function, it gives

$$\Lambda^k \| Ł^k(\mathbf{y} - \hat{\mathbf{y}}_k) \| \leq \Lambda^k \| R(Ł)^k(\mathbf{y} - \hat{\mathbf{y}}_k) \|.$$

Because the interpolant $\hat{\mathbf{y}}_k$ minimize the norm $\| R(Ł)^k \cdot \|$, we have

$$\| R(Ł)^k(\mathbf{y} - \hat{\mathbf{y}}_k) \| \leq \| R(Ł)^k\mathbf{y} \| + \| R(Ł)^k\hat{\mathbf{y}}_k \| \leq 2 \| R(Ł)^k\mathbf{y} \|.$$

As for functions $\mathbf{y} \in \mathrm{PW}_\omega(\mathcal{G}) \subset \mathrm{PW}_{R(\omega)}(\mathcal{G})$ the Bernstein inequality in Lemma 8 holds

$$\| R(Ł)^k\mathbf{y} \| \leq R(\omega)^k \| \mathbf{y} \|, k \in \mathbb{N}.$$

Putting everything together, we conclude the first part of Theorem 4:

$$\| \mathbf{y} - \hat{\mathbf{y}}_k \| \leq 2\Big(\Lambda R(\omega)\Big)^k \| \mathbf{y} \|, \Lambda R(\omega) < 1, k = 2^l, l \in \mathbb{N} \tag{21}$$

**Proof of the second part of the Theorem 4.**

Since $\Lambda R(\omega) < 1$ holds, it gives the following limit

$$\lim_{k \to \infty} (\Lambda R(\omega))^k = 0 \quad \text{and} \quad \lim_{k \to \infty} \| \mathbf{y} - \hat{\mathbf{y}}_k \| \leq 0$$

With the non-negativity of the norm, we have

$$\|\mathbf{y} - \hat{\mathbf{y}}_k\| \geq 0. \tag{22}$$

This implies the second part of the Theorem 4:

$$\mathbf{y} = \lim_{k \to \infty} \widetilde{\mathbf{y}}_k. \tag{23}$$

□

**Lemma 7 (restated from Lemma 4.1 in (Pesenson, 2009)).** *Suppose that Ł is a bounded self-adjoint positive definite operator in a Hilbert space $L_2(\mathcal{G})$, and $\| \phi \| \leq a \| Ł\phi \|$ holds true for any $\phi \in L_2(\mathcal{G})$ and a positive scalar $a > 0$, then for all $k = 2^l, l = 0, 1, \ldots$, the following inequality holds true*

$$\| \phi\| \leq a^k \|Ł^k\phi \| . \tag{24}$$

**Lemma 8 (restated from Theorem 2.1 in (Pesenson, 2008)).** *A function $\mathbf{f} \in L_2(\mathcal{G})$ belongs to $\mathrm{PW}_\omega(\mathcal{G})$ if and only if the following Bernstein inequality holds true for all $s \in \mathbb{R}_+$*

$$\| Ł^s\mathbf{y} \| \leq \omega^s \| \mathbf{y} \| . \tag{25}$$

### G.1 EXTRA DISCUSSION

In (Pesenson, 2008), the complementary set $S = \Omega_c = V - \Omega$ which admits Poincare inequality is called the $\Lambda$-set. Theorem 4 in our paper and Theorem 1.1 in (Pesenson, 2009) state that bandlimited functions $\mathbf{y} \in \mathrm{PW}_\omega$ can be reconstructed from their values on a uniqueness set $\Omega = V - S$. To better understand the concept of $\Lambda$-set, we restate Lemma 9 from (Pesenson, 2008) which presents the conditions for $\Lambda$-set. It is worth pointing out that (i) the second condition suggests that the vertices from $\Lambda$-set would likely be sparsely connected with the uniqueness set $\Omega$; and (ii) the vertices in $\Lambda$-set are disconnected with each other or isolated in the subgraph constructed by the vertices $S$, otherwise there always exists a non-zero function $\phi \in L_2(S), \| \phi \| \neq 0$ which makes $\| Ł\phi \| = 0$.

**Lemma 9** (**restated from Lemma 3.6 in (Pesenson, 2008)**). *Suppose that for a set of vertices $S \subset V$ (finite or infinite) the following holds true:*

1. *every point from $S$ is adjacent to a point from the boundary $bS$, the set of all vertices in $V$ which are not in $S$ but adjacent to a vertex in $S$;*

2. *for every $v \in S$ there exists at least one adjacent point $u_v \in bS$ whose adjacency set intersects $S$ only over $v$;*

3. *the number $\Lambda = \sup_{v \in s} d(v)$ is finite;*

*Then the set $S$ is a $\Lambda$-set which admits the Poincare inequality*

$$\| \phi \| \leq \Lambda \| Ł\phi \|, \phi \in L_2(S). \tag{26}$$

In our experiments for recommender systems, each user's ratings might not comply with Poincare inequality. This is because there exists some users who prefer niche products/movies (low-degree nodes). As shown in Fig. 2, user preferences on low-degree nodes are determined by high-frequency functions. When $R(\omega)$ is not large enough, Poincare inequality does not hold for such users. This also explains why our model performs poorly for cold items.

Regarding to choice of parameter $k$, empirical results show that using $k \geq 2$ does not help improve the performance, and note that when $k$ is large enough, all kernels will be reduced to bandlimited norm, i.e., $R(\lambda) = 1$ if $\lambda \leq \lambda_k \leq 1$, since the gap between eigenvalues shrinks.

## H PROOF OF THEOREM 5

*Proof.* Let $\boldsymbol{\xi}$ denote the random label noise which flips a 1 to 0 with rate $\rho$, assume that the sample $\mathbf{s} = \mathbf{y} + \boldsymbol{\xi}$ is observed from $\mathbf{y}$ under noise $\boldsymbol{\xi}$, then for a graph spectral filter $\mathbf{H}_\varphi = (\mathbf{I} + R(Ł)/\varphi)^{-1}$ with positive $\varphi > 0$, we have

$$\mathbf{E}\Big[\mathrm{MSE}(\mathbf{y}, \hat{\mathbf{y}})\Big] = \frac{1}{n}\mathbf{E} \| \mathbf{y} - \mathbf{H}_\varphi(\mathbf{y} + \boldsymbol{\xi}) \|^2$$

$$\leq \frac{1}{n}\mathbf{E} \| \mathbf{H}_\varphi\boldsymbol{\xi} \|^2 + \frac{1}{n} \| (\mathbf{I} - \mathbf{H}_\varphi)\mathbf{y} \|^2, \tag{27}$$

where the last inequality holds due to the triangular property of matrix norm.

To bound $\mathbf{E} \| \mathbf{H}_\varphi\boldsymbol{\xi} \|^2$, let $C_n = R^{1/2}(\omega) \| \mathbf{y} \|$, then

$$\mathbf{E} \| \mathbf{H}_\varphi\boldsymbol{\xi} \|^2 \stackrel{(a)}{=} \sum_{\mathbf{y}(v)=1} \rho(\mathbf{H}_{\varphi,(*,v)} \times -1)^2 + (1 - \rho)(\mathbf{H}_{\varphi,(*,v)} \times 0)^2$$

$$= \rho \sum_{\mathbf{y}(v)=1} (\mathbf{H}_{\varphi,(*,v)}\mathbf{y}(v))^2 = \rho \| \mathbf{H}_\varphi\mathbf{y} \|^2$$

$$\stackrel{(b)}{\leq} \sup_{\|R^{1/2}(Ł)\mathbf{y}\| \leq C_n} \rho \| \mathbf{H}_\varphi\mathbf{y} \|^2 = \sup_{\|\mathbf{z}\| \leq C_n} \rho \| \mathbf{H}_\varphi R^{-1/2}(Ł)\mathbf{z} \|^2$$

$$= \rho C_n^2 \sigma_{\max}^2\Big(\mathbf{H}_\varphi R^{-1/2}(Ł)\Big) = \rho C_n^2 \max_{l=1,\dots,n} \frac{1}{(1 + R(\lambda_l)/\varphi)^2} \frac{1}{R(\lambda_l)}$$

$$\leq \frac{\rho\varphi^2 C_n^2}{R(\lambda_1)(\varphi + R(\lambda_1))^2}, \tag{28}$$

where (a) follows the definition of the flip random noise $\boldsymbol{\xi}$ and (b) holds to the fact that $\mathbf{y}$ is in the Paley-Wiener space $\mathrm{PW}_\omega(\mathcal{G})$. As for the second term,

$$
\begin{aligned}
\| (\mathbf{I} - \mathbf{H}_\varphi)\mathbf{y} \|^2 &\leq \sup_{\|R^{1/2}(\text{Ł})\mathbf{y}\| \leq C_n} \| (\mathbf{I} - \mathbf{H}_\varphi)\mathbf{y} \|^2 \\
&\overset{(a)}{=} \sup_{\|\mathbf{z}\| \leq C_n} \| (\mathbf{I} - \mathbf{H}_\varphi)R^{-1/2}(\text{Ł})\mathbf{z} \|^2 \\
&= C_n^2 \sigma_{\max}^2\Big((\mathbf{I} - \mathbf{H}_\varphi)R^{-1/2}(\text{Ł})\Big) \\
&= C_n^2 \max_{l=1,\dots,n} \Big(1 - \frac{1}{1 + R(\lambda_l)/\varphi}\Big)^2 \frac{1}{R(\lambda_l)} \\
&= \frac{C_n^2}{\varphi} \max_{l=1,\dots,n} \frac{R(\lambda_l)/\varphi}{(R(\lambda_l)/\varphi + 1)^2} \\
&\overset{(b)}{\leq} \frac{C_n^2}{4\varphi}.
\end{aligned}
\tag{29}
$$

where (a) holds due to the fact that the eigenvectors of $\mathbf{I} - \mathbf{H}_\varphi$ are the eigenvectors of $R(\text{Ł})$; and (b) follows the simple upper bound $x/(1+x)^2 \leq 1/4$ for $x \geq 0$.

By combing everything together, we conclude the result

$$
\mathbf{E}\Big[\mathrm{MSE}(\mathbf{y}, \hat{\mathbf{y}})\Big] \leq \frac{C_n^2}{n}\Big(\frac{\rho\varphi^2}{R(\lambda_1)(\varphi + R(\lambda_1))^2} + \frac{1}{4\varphi}\Big).
\tag{30}
$$

$\square$

### H.1 EXTRA DISCUSSION

Choosing $\varphi$ to balance the two terms on the right-hand side above gives $\varphi^* = \infty$ for $\rho < 1/8$ and $1 + R(\lambda_1)/\varphi^* = 2\rho^{1/3}$ for $\rho \geq 1/8$. Plugging in this choice, we have the upper bound if $\rho \geq \frac{1}{8}$

$$
\mathbf{E}\Big[\mathrm{MSE}(\mathbf{y}, \hat{\mathbf{y}})\Big] \leq \frac{C_n^2}{4R(\lambda_1)n}(3\rho^{1/3} - 1),
\tag{31}
$$

and if $\rho < \frac{1}{8}$, then the upper bound is

$$
\mathbf{E}\Big[\mathrm{MSE}(\mathbf{y}, \hat{\mathbf{y}})\Big] \leq \frac{C_n^2\rho}{4R(\lambda_1)n}.
\tag{32}
$$

This result implies that we can use a large $\varphi$ to obtain accurate reconstruction when the flip rate $\rho$ is not greater than $1/8$, and $\varphi$ need to be carefully tuned when the flip rate $\rho$ is greater than $1/8$.

## I PROOF OF PROPOSITION 6

As below we present the proof in a Bayesian framework, and the reader is referred to (Maybeck, 1982) for a geometrical interpretation of Monte Carlo estimate statistics.

**Proof of the minimal variance**

To minimize the estimate variance, we need to minimize the main diagonal of the covariance $\mathbf{P}_{\mathrm{new}}$:

$$
\mathrm{trace}\Big(\mathbf{P}_{\mathrm{new}}\Big) = \mathrm{trace}\Big((\mathbf{I} - \mathbf{K})\bar{\mathbf{P}}_{\mathrm{new}}(\mathbf{I} - \mathbf{K})^\top + \mathbf{K}\boldsymbol{\Sigma}_\mu\mathbf{K}^\top\Big).
$$

Then, we differentiate the trace of $\mathbf{P}_{\mathrm{new}}$ with respect to $\mathbf{K}$

$$
\frac{\mathrm{d}\,\mathrm{trace}\Big(\mathbf{P}_{\mathrm{new}}\Big)}{\mathrm{d}\,\mathbf{K}} = \mathrm{trace}\Big(2\mathbf{K}\bar{\mathbf{P}}_{\mathrm{new}} - 2\bar{\mathbf{P}}_{\mathrm{new}}\Big) + \mathrm{trace}\Big(2\mathbf{K}\boldsymbol{\Sigma}_u\Big).
$$

The optimal $\mathbf{K}$ which minimizes the variance should satisfy $\mathrm{d}\,\mathrm{trace}(\mathbf{P}_{\mathrm{new}})/\mathrm{d}\,\mathbf{K} = 0$, then it gives

$$
\mathbf{K}(\mathbf{I} + \bar{\mathbf{P}}_{\mathrm{new}}) = \bar{\mathbf{P}}_{\mathrm{new}}.
$$

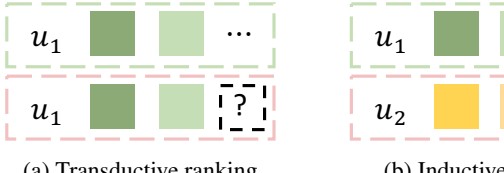

|   |   |
|---|---|
| (a) Transductive ranking | (b) Inductive ranking |

Figure 5: Evaluation protocols, where the users in top block (green) are used for training and the ones in bottom block (pink) are used for evaluation. (a) transductive ranking, where the model performance is evaluated based on the users already known during the model training; (b) inductive ranking, the model performance is evaluated using the users unseen during the model training.

This implies that the variance of estimate $\hat{\mathbf{x}}_{\text{new}}$ is minimized when $\mathbf{K} = \bar{\mathbf{P}}_{\text{new}}(\mathbf{I} + \bar{\mathbf{P}}_{\text{new}})^{-}$.

**Proof of the unbiasedness**

Suppose that the obsolete estimate $\hat{\mathbf{x}}$ is unbiased, i.e. $\mathbb{E}\hat{\mathbf{x}} = \mathbf{x}$, then using Eq. (11) we have

$$\mathbf{E}\big(\bar{\mathbf{x}}_{\text{new}}\big) = \mathbf{E}\big(\hat{\mathbf{x}} + \mathbf{F}\Delta\mathbf{s}\big) = \mathbf{x} + \mathbf{F}\Delta\mathbf{s} = \mathbf{x}_{\text{new}}.$$

Because of Eq. (12) and the measurement noise $\nu$ has zero mean, it gives

$$\mathbf{E}\big(\mathbf{z}_{\text{new}}\big) = \mathbf{E}\big(\mathbf{x}_{\text{new}} + \nu\big) = \mathbf{x}_{\text{new}}.$$

Putting everything together, we conclude the following result

$$\mathbf{E}\big(\hat{\mathbf{x}}_{\text{new}}\big) = \mathbf{E}\big(\bar{\mathbf{x}}_{\text{new}} + \mathbf{K}(\mathbf{z}_{\text{new}} - \bar{\mathbf{x}}_{\text{new}})\big) = \mathbf{x}_{\text{new}} + \mathbf{K}(\mathbf{x}_{\text{new}} - \mathbf{x}_{\text{new}}) = \mathbf{x}_{\text{new}}. \tag{33}$$

This implies that the estimate state $\hat{\mathbf{x}}_{\text{new}}$ is unbiased.

## J  IMPLEMENTATION DETAILS

In this section, we present the details for our implementation in Section 5 including the additional dataset details, evaluation protocols, model architectures in order for reproducibility. All the experiments are conducted on the machines with Xeon 3175X CPU, 128G memory and P40 GPU with 24 GB memory. The configurations of our environments and packages are listed below:

- Ubuntu 16.04
- CUDA 10.2
- Python 3.7
- Tensorflow 1.15.3
- Pytorch 1.10
- DGL 0.7.1
- NumPy 1.19.0 with MKL Intel

### J.1  ADDITIONAL DATASET DETAILS

We use three real-world datasets which are processed in line with (Liang et al., 2018; Steck, 2019): (1) for Koubei[2], we keep users with at least 5 records and items that have been purchased by at least 100 users; and (2) for Tmall[3], we keep users who click at least 10 items and items which have been seen by at least 200 users; and (3) for Netflix[4], we keep all of the users and items. In addition, we chose the random seed as 9876 when splitting the users into training/validation/test sets.

---

[2]https://tianchi.aliyun.com/dataset/dataDetail?dataId=53
[3]https://tianchi.aliyun.com/dataset/dataDetail?dataId=35680
[4]https://kaggle.com/netflix-inc/netflix-prize-data

## J.2 EVALUATION PROTOCOLS

In Figure 5, we illustrate the difference between the transductive ranking and inductive ranking evaluation protocols. In the transductive ranking problem, the model performance is evaluated on the users already known during the model training, whereas the model performance is evaluated on the unseen users in the inductive ranking problems. It is worth noting that in the testing phrase, we sort all interactions of the validation/test users in chronological order, holding out the last one interaction for testing and inductively generating necessary representations on the rest data. In a nutshell, we evaluate our approach and the baselines for the challenging inductive next-item prediction problem.

## J.3 EVALUATION METRICS

We adopt hit-rate (HR) and normalized discounted cumulative gain (NDCG) to evaluate the model performance. Suppose that the model provide $N$ recommended items for user $u$ as $R_u$, let $T_u$ denote the interacted items of the user, then HR is computed as follows:

$$\text{HR@N} = \mathbf{E}_u \ \mathbf{1}|T_u \cap R_u| \tag{34}$$

where $\mathbf{1}|\Omega|$ is equal to 1 if set $\Omega$ is not empty and is equal to 0 otherwise. NDCG evaluates ranking performance by taking the positions of correct items into consideration:

$$\text{NDCG@N} = \frac{1}{Z}\text{DCG@N} = \frac{1}{Z}\sum_{j=1}^{N}\frac{2^{\mathbf{1}|R_u^j \cap T_u|} - 1}{\log_2(j+1)} \tag{35}$$

where $Z$ is the normalized constant that represents the maximum values of DCG@N for $T_u$.

## J.4 GRAPH LAPLACIAN

Let $\mathbf{R}$ denote the item-user rating matrix, $\mathbf{D}_v$ and $\mathbf{D}_e$ denotes the diagonal degree matrix of vertices and edges respectively, then graph Laplacian matrix used in our experiments is defined as follows:

$$\text{Ł} = \mathbf{I} - \mathbf{D}_v^{-1/2}\mathbf{R}\mathbf{D}_e^{-}\mathbf{R}^{\top}\mathbf{D}_v^{-1/2}. \tag{36}$$

where $\mathbf{I}$ is identity matrix.

## J.5 DISCUSSION ON PREDICTION FUNCTIONS

In experiments, we focus on making personalized recommendations to the users, so that we are interested in the ranks of the items for each user. Specifically, for top-k ranking problem we choose the items with the $k$-largest predicted ratings,

$$\text{Recommendation@k} = \max_{|O|=k}\sum_{v \in O, v \notin \Omega_+}\mathbf{y}(v). \tag{37}$$

More importantly, our proposed method is also suitable for the link prediction problem, where the goal is classify whether an edge between two vertices exists or not. This can be done by choosing a splitting point to partition the candidate edges into two parts. There are many different ways of choosing such splitting point. One can select the optimal splitting point based on the ROC or AUC results on the validation set.

## J.6 MODEL ARCHITECTURES

As mentioned before, we equip IDCF (Wu et al., 2021) with different GNN architectures as the backbone. Here we introduce the details for them.

**GAT.** We use the *GATConv* layer available in DGL for implementation. The detailed architecture description is as below:

- A sequence of one-layer *GATConv* with four heads.
- Add self-loop and use batch normalization for graph convolution in each layer.

- Use *tanh* as the activation.
- Use inner product between user embedding and item embedding as ranking score.

**GraphSAGE.** We use the *SAGEConv* layer available in DGL for implementation. The detailed architecture description is as below:

- A sequence of two-layer *SAGEConv*.
- Add self-loop and use batch normalization for graph convolution in each layer.
- Use *ReLU* as the activation.
- Use inner product between user embedding and item embedding as ranking score.

**SGC.** We use the *SGConv* layer available in DGL for implementation. The detailed architecture description is as below:

- One-layer *SGConv* with two hops.
- Add self-loop and use batch normalization for graph convolution in each layer.
- Use *ReLU* as the activation.
- Use inner product between user embedding and item embedding as ranking score.

**ChebyNet.** We use the *ChebConv* layer available in DGL for implementation. The detailed architecture description is as below:

- One-layer *ChebConv* with two hops.
- Add self-loop and use batch normalization for graph convolution in each layer.
- Use *ReLU* as the activation.
- Use inner product between user embedding and item embedding as ranking score.

**ARMA.** We use the *ARMAConv* layer available in DGL for implementation. The detailed architecture description is as below:

- One-layer *ARMAConv* with two hops.
- Add self-loop and use batch normalization for graph convolution in each layer.
- Use *tanh* as the activation.
- Use inner product between user embedding and item embedding as ranking score.

**We also summarize the implementation details of the compared sequential baselines as follows.**

**SASREC.**[5] We use the software provided by the authors for experiments. The detailed architecture description is as below:

- A sequence of two-block Transformer with one head.
- Use maximum sequence length to 30.
- Use inner product between user embedding and item embedding as ranking score.

**BERT4REC.**[6] We use the software provided by the authors for experiments. The detailed architecture description is as below:

- A sequence of three-block Transformer with eight heads.
- Use maximum sequence length to 30 with the masked probability 0.2.
- Use inner product between user embedding and item embedding as ranking score.

---

[5] https://github.com/kang205/SASRec
[6] https://github.com/FeiSun/BERT4Rec

**GREC.**[7] We use the software provided by the authors for experiments. The detailed architecture description is as below:

- A sequence of six-layer dilated CNN with degree $1, 2, 2, 4, 4, 8$.
- Use maximum sequence length to $30$ with the masked probability $0.2$.
- Use inner product between user embedding and item embedding as ranking score.

---

[7]https://github.com/fajieyuan/WWW2020-grec

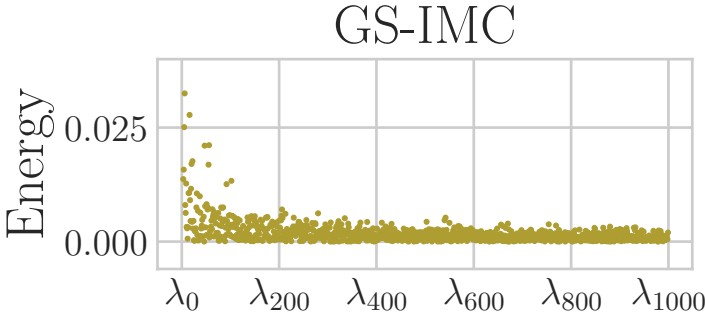

Figure 6: Spectrum analysis for static GS-IMC, of which the energy is concentrated on the low frequencies since the high-frequency functions are highly penalized during minimization.

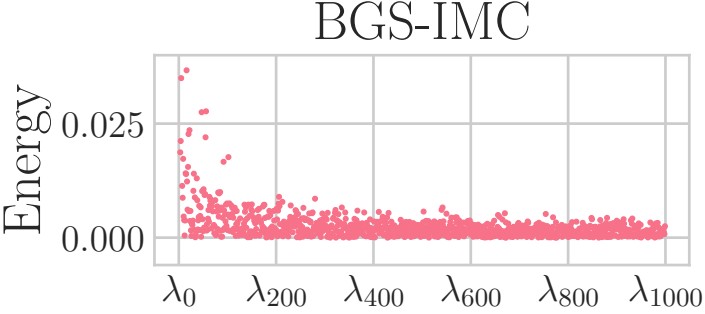

Figure 7: Spectrum analysis for static BGS-IMC, of which the energy is concentrated on the low frequencies since the high-frequency functions are highly penalized during minimization.

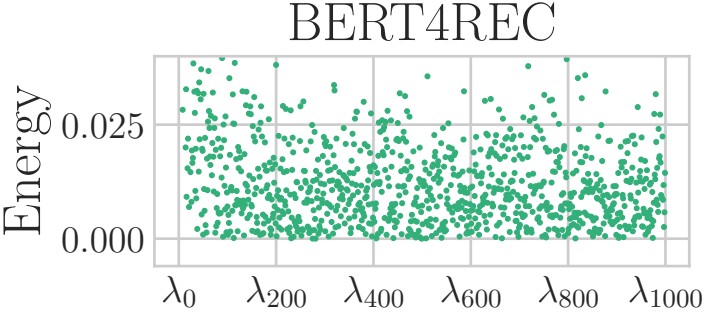

Figure 8: Spectrum analysis for sequential BERT4REC, of which the high-frequency functions are not highly penalized and can aggressively raise the rankings of unpopular items .

