# OpenReview forum: "Graph Signal Sampling for Inductive One-Bit Matrix Completion: a Closed-form Solution"
_ICLR.cc/2023/Conference — ICLR 2023 poster_

### Official Review · Reviewer_ZrCo · 2022-10-23

**Confidence:** 3
**Correctness:** 4
**Technical Novelty And Significance:** 3
**Empirical Novelty And Significance:** 2
**Recommendation:** 8

**Clarity, Quality, Novelty And Reproducibility:**

**Novelty**: the GS-IMC algorithm seems to be a straightforward generalization of SGMC (Chen et al.), with very similar performance. There are however some novel consistency results (one on the effect of bandwidth limiting, and the other on the effect of noise). To the best of my knowledge, the BGS-IMC is completely novel in one-bit completion, and is the main innovation of the paper.

**Clarity/Reproducibility**: here are some more precise remarks on the clarity issues
- Model definition: the presentation at the beginning of the paper is very confusing; the link between $M, R, y$ and $s$ is hard to grasp, and would benefit from a much more formal presentation similar to the one of graph sampling. In particular, the noise model is unclear : are bits only flipped from 1 to 0, or can there be opposite flips ?
- BGS-IMC : the setup is again slightly confusing. I don't exactly understand why you introduce two variables $x$ and $z$, when the difference between them is simply some additive noise; in general I feel like Equations (11) and (12) need more explanation. On the other hand, once we accept the model, the rest is fairly straightforward and explained well.
- Experiments: again, the experimental setup may need more explanation; the metrics in Tables 3 and 4 are not defined, especially the `@50` or `@100` suffixes. I also didn't see which regularizer $R$ was used to produce the results in the tables. However, I really appreciate the effort to made the algorithms compared against as efficient as possible.

*Minor remarks**:
- Figure 2 (and 4-6): the algorithm names are wrong
- throughout the paper (mainly in section 4, see e.g. Eq. 18 and below Eq. 15), $A^-$ instead of $A^{-1}$ is used for inverting matrices; or is that another operation ?

**Strength And Weaknesses:**

The main strengths of the proposed algorithms are their scalability and explainability: since they are fairly simple to implement (the first one even has a closed-form solution), they are much faster than heavier neural network counterparts; and on the other hand even the learned elements of the algorithms (such as the covariance matrices of BGS-IMC) can have interesting interpretations. The algorithms are quite thoroughly studied, with ablation and scalability studies performed in the appendix.

On the other hand, the paper is very hermetic for people that are not familiar with the setting and state of the art; it took me quite a long time to grasp the exact setting of the original problem. The introduction to BGS-IMC suffers the same problem : it is hard to understand exactly what $\Delta s$ is, and why the model is setup that way. Overall, this paper needs significant work to make it accessible beyond non-experts of the field.

**Summary Of The Paper:**

This paper studies the problem of one-bit completion: the task is to recover a vector $y \in \\{0, 1\\}^n$ from the observation of a subset of its positive entries. The model assume that there exists a matrix $R \in \\{0, 1\\}^{m \times n}$ containing additional information about the model, e.g. $m$ previous partial observations of similar data.

The main idea is that this matrix $R$ can be turned into a graph on $n$ vertices, whose weights represent the similarity between vertices. The completion problem can thus be cast as a graph signal sampling problem using the laplacian $L$ of the graph: the general assumption is that $y$ is mostly concentrated on the low-frequency eigenvectors of $L$.

Next, the authors study the on-line version of this problem, where the observations can be noisy. In this case, they propose a Bayesian algorithm that assumes the noise is Gaussian in the Fourier basis of $L$. This is an adaptation of prediction-correction-update algorithms to the case of (0, 1) observations.

Finally, they propose several experiments to gauge the performance of their algorithms. The first part compares it to other graph-based methods, and the second part to more complicated (e.g. transformer-based) sequential recommendation algorithms. In the first case, the Bayesian version BGS-IMC is shown to mostly outperform its counterparts, while in the second case, the performance is comparable (sometimes slightly worse) than other architectures but the runtime is much lower.

**Summary Of The Review:**

This is a very interesting paper, that introduces a framework for one-bit completion that's both efficient and comparable to SOTA accuracy. However, it is for now not accessible beyond a very small subset of experts, and needs clarity changes to broaden its appeal.

---

> ### Author Response · Authors · 2022-11-13
> **Thank you so much for valuable comments**
>
> We thank the reviewer for the efforts and time in reviewing our paper. Also, we appreciate the reviewer's recognition of our novelty and contribution. We provided detailed responses to the reviewer's comments/questions as follows.
>
> **Q**:  Clarity.
>
> **A**: We have polished the paper to clarify the notations and the setup of both GS-IMC and BGS-IMC.
>
> Specifically, we have added figures to the introduction section, where we take recommender system as an example to illustrate the relationships among $M, R, y, s$ as well as the noise $\xi$. Regarding to the assumption that $\xi$ only flips 1s to 0s, we believe that it is well suited for real world applications such as recommender systems and social networks. On the other hand, BGS-IMC considers the measurement noise, which implicitly takes into account the uncertainty in positive data.
>
> We have also added figures to BGS-IMC section, where we emphasize that the measurement $z$ represents the prediction $\hat{y}$ in the graph Fourier domain, and $z$ follows a stochastic process where hidden state $x$ determines the measurement in the presence of noise $\nu$. This address the uncertainty in positive examples, as discussed previously. The challenges lie in the choice of state transition equation: (1) the model uncertainty needs to be considered; (2) the state transition needs to represent the evolution of model predictions (i.e., Eq. 10). One feasible solution is to directly multiply both sides of Eq. 10 by Fourier basis $U$, the resulting equation is Eq. 11 which essentially follows Eq. 10 in the graph Fourier domain. The additional noise $\eta$ models the process noise, which determines the weight of the observed measurement. One more interesting observation is that $\eta$ has the ability to control the difference between GS-IMC and BGS-IMC. For example, when the model uncertainty (i.e., the determinant of $\Sigma_\eta$) is high, BGS-IMC will be reduced to GS-IMC.
>
> **Q**: Reproducibility.
>
> **A**: We have added Appendix J.3 to describe the evaluation metrics. Thanks for the advice.
>
> We report the hit-rate and NDCG results in Table 3 and Table 4, respectively. To avoid the misunderstanding, we replace @50 by H@50 in Table 3 and by N@50 in Table 4. Note that we report the best result we can have, for example, hit-rate (Table 2) for BGS-IMC on Koubei corresponds to BGS-IMC-random walk (in Table 4), hit-rate (Table 2) for BGS-IMC on Netflix corresponds to BGS-IMC-Tikhonov (in Table 4).
>
> **Q**: Typos.
>
> **A**:  We have revised the paper with right model names and consistent notation of matrix inverse, i.e., $A^-$. Thank you so much.

---

> > ### Comment · Reviewer_ZrCo · 2022-11-28
> > **Score raised**
> >
> > Thank you for the improvements to the paper structure, it is indeed easier to follow in the new version. Accordingly, I have raised my score.

---

> > > ### Author Response · Authors · 2022-11-28
> > > **Thank you for raising the score**
> > >
> > > Dear Reviewer **ZrCo**,
> > >
> > > Many thanks for all the constructive comments and very positive evaluations. We really appreciate reviewer **ZrCo** for increasing our score.
> > >
> > > We are again very thankful for your time and support!
> > >
> > > Best wishes,
> > >
> > > Authors

---

### Official Review · Reviewer_fzvN · 2022-10-24

**Confidence:** 4
**Clarity, Quality, Novelty And Reproducibility:** see above
**Correctness:** 2
**Technical Novelty And Significance:** 2
**Empirical Novelty And Significance:** 2
**Recommendation:** 3

**Strength And Weaknesses:**

Strength: Authors consider a useful application of graph signal processing to recommender systems.

Weaknesses:
* The main contributions of the paper are unclear. Is it a novel graph signal model ? Is it a novel graph signal recovery method (Eq. (5)) ?

* At least in its application to recommender systems I would like to see a comparison of the proposed methods with existing network Lasso methods as proposed e.g. in

N. Tran, H. Ambos and A. Jung, "Classifying Partially Labeled Networked Data VIA Logistic Network Lasso," ICASSP 2020 - 2020 IEEE International Conference on Acoustics, Speech and Signal Processing (ICASSP), 2020, pp. 3832-3836, doi: 10.1109/ICASSP40776.2020.9054408.

A. Jung, "Networked Exponential Families for Big Data Over Networks," in IEEE Access, vol. 8, pp. 202897-202909, 2020, doi: 10.1109/ACCESS.2020.3033817.

* Theorem 3 needs more discussion. How to choose the parameter k ? How restrictive is the Poincare condition ? Is this condition satisfied in the numerical experiments ?

* The problem formulation needs to be made more precise and placed earlier in the paper. Currently it seems only described in Section 4.1. which is too late.

* What is \hat{y} in (9) ?

* Pls explain more clearly the probability distribution underlying the expectation in Eq. (9).

* Pls discuss more explicitly how the graph (Laplacian) has been obtained for the numerical experiments.

* "The classical sampling theorem states that functions..." what is the "classical sampling theorem" ?


minor errors:

* "...the testing phrase,.."

* "..experiments, we define the hypergraph using matrix R.." unclear what the matrix R is/how obtained.

* "..methods are not well suited to our 1-bit matrix completion problem due to the issues of 1-bit quantization..." pls try to be more specific. what are these "issues" ?



**Summary Of The Paper:**

Authors extend graph signal processing techniques to cope with discrete random label noise.


**Summary Of The Review:**

see above

---

> ### Author Response · Authors · 2022-11-13
> **Clarify the misunderstandings**
>
> We thank the reviewer for the thorough review. In the following, we provided detailed responses to the reviewer’s comments/questions.
>
> **Q**: Novelty and Clarity
>
> **A**: We believe that there exists a fundamental misunderstanding, due to our condense presentation, which makes the reviewer underestimate our technical novelty and contribution.
>
> In this paper, we propose a new inductive (1-bit) matrix completion framework which enjoys benefits of graph signal processing techniques. Specifically, we develop a new closed-form approach, entitled GS-IMC (i.e., Eq. 5-7), to efficiently recover unseen column from their observations that are present at test time. Or in context of recommender systems, GS-IMC has the ability to serve new users, who are not exposed to the model during training but appear at testing stage.
>
> Additionally, we extend to a new Bayesian GS-IMC, called BGS-IMC (i.e., Eq. 11-18), which considers the online learning scenarios. Compared to GS-IMC, BGS-IMC takes into account the measurement uncertainty and the model uncertainty, which is more realistic in real world applications. Further, we develop a prediction-correction update algorithm to obtain unbiased and variance-minimal estimator.
>
> Overall, Section 2 provides necessary background and preliminary. Section 3 describes our new GS-IMC approach with error analysis. Section 4 proposes BGS-IMC to fulfill the requirements of online learning scenario.

---

> > ### Comment · Reviewer_fzvN · 2022-12-04
> > **Contributions**
> >
> > - "we propose a new inductive (1-bit) matrix completion framework "  what exactly is a "framework"?
> > - "..., to efficiently recover unseen column from their observations that are present at test time." in which precise sense is your method efficient ? does the estimation error of your method match fundamental lower bounds that cannot be overcome by any method ?

---

> > > ### Author Response · Authors · 2022-12-04
> > > **Responses to Reviewer fzvN**
> > >
> > > Dear Reviewer **fzvN**,
> > >
> > > We sincerely thank reviewer **fzvN** for the valuable comments and the positive support.
> > >
> > > **Q**: What is a framework?
> > >
> > > **A**:We are sorry for the caused confusion to you. We put it more specifically here and hope it could ease your concern.
> > >
> > > To better understand our GS-IMC framework (Section 3), we first clarify the research problem focused by this paper. Let us take a toy example in the context of recommender system as shown in Figure. 1b.
> > >
> > >
> > > > **Problem**: Assume that we observe an item-user rating matrix $R$ of size $4\times{2}$, we call
> > > > - the 0-th male user, Bob with ratings $R_{*,0}$,
> > > > - the 1-st female user, Alice with ratings $R_{*,1}$.
> > > >
> > > > Specifically, we observed that Bob bought t-shirt and jean, while Alice bought dress and skirt.
> > > >
> > > >**Goal**: At testing stage, a *new* user Charlie comes to the system and bought a skirt, then our goal is to recommend other products to him.
> > >
> > > In this case, Charlie represents the new column out of the observed matrix $R$, and we have the observation Charlie(skirt)=1, consisting only of ones. The goal of this paper is to recover Charlie's missing data (e.g., Charlie(dress)=? ) from his observation (i.e., Charlie(skirt)=1) that is present at testing stage.
> > >
> > > To solve the problem, we use the item-user matrix $R$ to build an item-item graph $G$. For example, we define the Laplacian matrix of the item-item graph as $L= I - D^{-0.5}R{R}^TD^{-0.5}$. Matrix D denotes the degree matrix.
> > >
> > > > **GS-IMC Framework**:
> > > >
> > > >  **Step1: Graph Signal Treatment**. we regard the purchase history of Charlie as a graph signal, a function from the vertex space (i.e., T-Shirt, Jean, Dress, Skirt) to the binary space (i.e., 0 or 1).
> > > > - So far, we observe the function value, i.e., Charlie(skirt) = 1
> > > >
> > > >  **Step2: Graph Signal Sampling**. Recovering Charlie's missing data is equivalent to reconstructing a function on graph. To avoid degenerate solutions, we sample the product "Jean" as a negative item, then we solve the problem as follows:
> > > >
> > > > $\quad\quad\min_f \parallel Reg(L)f \parallel$
> > > >
> > > > s.t.,
> > > >
> > > > $\quad\quad[f$(skirt) - 1$]^2$ + $[f$ (Jean) - 0$]^2 \le{\epsilon}$
> > > >
> > > > where $Reg(L)$ represents a regularized kernel. We provide examples in Table 1.
> > >
> > > **Why it is a framework?** The reason is that when we adopt a different regularized kernel, we will have a different model.
> > >
> > > For example, as shown in Appendix. B,
> > >
> > > - When we adopt the bandlimited norm, [4] is a special case of our framework;
> > > - When we adopt the one-step random walk norm, [5] is a special case of our framework;
> > >
> > > [4] Scalable and explainable 1-bit matrix completion via graph signal learning. AAAI 2021.
> > >
> > > [5] Markov random fields for collaborative filtering. NeurIPS 2019.
> > >
> > >
> > > **Why TV appraoch fails in our setting?** We recall that our observed data consists only of positive ones, and let us see the definition of typical TV approach:
> > >
> > > > $\quad\quad \min_f \parallel Lf \parallel_1$
> > > >
> > > > s.t.,
> > > >
> > > >$\quad\quad [f$(skirt$) - 1]^2 \le \epsilon$
> > >
> > > In our toy example, we only observe Charlie(skirt) =1 and nothing else. We see that the optimal solution is constant $f=1$ as it achieves zero loss, i.e.,
> > > - $\parallel L\mathbf{1} \parallel_1=0$
> > > - $ [f$(skirt$) - 1]^2=[1-1]=0$
> > >
> > > In a nutshell, typical TV approach provides a degenerate solution.
> > >
> > > **Lower bound comparisons.** We believe that our problem setting is different from previous works.
> > > 1. We deal with positive-unlabelled data, and typical TV approaches do not work as we discussed;
> > > 2. We do not assume any content information, so we cannot learn a classifier that maps node features to binary labels (we will discuss later);
> > >
> > > To the best of our knowledge, we are the first to provide the reconstruction error in this particular problem setting. Note that the most closely related works [4,5] do not provide any theory results.
> > >
> > >
> > > **Q2**: The relationships between Efficiency and Accuracy.
> > >
> > > **A**: This is a very interesting question.
> > >
> > > We first admit that the closed-form solution of GS-IMC requires to perform the eigenvalue decomposition,
> > > > which takes $O(n^3)$ times, cubic to the number of nodes n (i.e., #items in the toy example).
> > >
> > > This is not acceptable for large graphs, and therefore we present an approximation algorithm in Appendix. D,
> > > > which takes $O(lnk+k^3)$, linear to the number of nodes n, where $k$ is the number of desired eigenvectors.
> > >
> > > As shown in Table 7, the prediction accuracy does not drop significantly, while the computational time is reduced from 12 minutes to 1.5 minutes on the largest Netflix data. Note that ChebyNet consumes 598 minutes.
> > >
> > > Overall, as shown in Table 2-3 and 7, our approach outperforms all baslines while consuming only a small fraction of training time required by baselines.
> > >
> > >
> > > We are again very thankful for your extra time of reviewing and the suggestion! We will adjust it in the revision.
> > >
> > > Best wishes,
> > >
> > > Authors

---

> ### Author Response · Authors · 2022-11-13
> **Detailed Feedback**
>
> **Q**: Comparison to Network Lasso.
>
> **A**: As discussed in our main paper, total variation denoising [1,2] is orthogonal to us, because these approaches, including Jung’s work, deal with real-valued data, while our methods are proposed to cope with positive-unlabeled data. We believe that our problem setting is more complicated, since the unlabeled data is a mixture of unobserved positive examples and true negative examples. In addition, existing theories (as well as Theorem 1 in [3]) analyze the recoverability considering statistical noise to be continuous Gaussian. By contrast, in 1-bit matrix completion, we study upper bounds in the presence of discrete random label noise that flips 1s to 0s.
>
> To make our experiments complete, we adopt the program (https://github.com/alexjungaalto/nLassoExpFamPDSimulations) provided by the authors and modify their **MATLAB** codes for top-N ranking. It takes tremendous memory when running on the Netflix prize dataset. We borrowed a Window workstation (MATLAB is not friendly to our Linux workstation) and managed to run the program. However, the original model recommends same items to all testing users (trivial solution takes very little time). To avoid this, we further introduce negative sampling techniques but this model relies on proximal gradient techniques to optimize the objective function. It takes a much longer time to achieve meaningful solutions, and the program does not seem to complete even after three days.
>
> [1] Total variation classes beyond 1d: Minimax rates, and the limitations of linear smoothers. NIPS 2016.
>
> [2] Higher-order total variation classes on grids: Minimax theory and trend filtering method. NeurIPS 2017.
>
> [3] Networked Exponential Families for Big Data Over Networks. IEEE Access.
>
>
> **Q**: More discussions on Theorem 3.
>
> **A**: We have added **Appendix G.1** to discuss more about the choice of $k$ and Poincare condition.
>
> Regarding to $k$, empirical results show that using $k\ge{2}$ does not significantly improve the accuracy, and we note that when $k$ is large enough, the gap between different eigenvalues will become small. In this case, all kernels will converge to bandlimited norm, i.e., $R(\lambda)=1$ for $\lambda\le\omega$.
>
> Regarding to Poincare condition, we believe that it might not hold for the users who prefer niche movies/products. Fig. 1 shows that low-frequency signal represents user preference on popular items, while high-frequency signal reflects user preference on cold items (i.e., low degree vertices). From this viewpoint, when $R(\omega)$ is not sufficiently large, Poincare condition will not hold for the users who prefer niche items. This also explains why our GS-IMC model performs poorly for cold items (see Fig. 1, the recall for cold items is as low as 0.03, while the recall for popular items is up to 0.38). However, we argue that large $R(\omega)$ leads to improved accuracy while requiring more computational resources. There is a trade-off between the model accuracy and the system efficiency.
>
> **Q**: More discussions on Theorem 4.
>
> **A**: Theorem 4 studies the risk of $\hat{y}$ (i.e., solution in Eq. 7) in the expectation of its MSE. In recommender systems, we believe it is reasonable to assume that $y_a, y_b$ for user a and b are uniformly and independently distributed. In such case, we can use Hoeffding inequality to derive a new bound with statistical confidence. However, this assumption might not hold in some domains, and thus we choose to not specify the distribution of random variable $y$.
>
> **Q**: Implementation details.
>
> **A**: We add more implementation details in **Appendix J**, including the definition of graph Laplacian and evaluation metrics. In our experiments, we adopt hypergraph Laplacian $I-D_v^{-.5}RD_e^-R D_v^{-.5}$ which has been summarized in Section 3.1.
>
> Also, we explicitly state the issue of 1-bit quantization, i.e., existing graph signal sampling approaches yield degenerate solutions when applying them to 1-bit matrix completion. Such issue has been discussed in the introduction section and also emphasized in Definition 3 (graph signal sampling).

---

> > ### Comment · Reviewer_fzvN · 2022-12-04
> > **Networked Exponential Families. Poincare Condition.**
> >
> > Note that the methods in [3] allow the local datasets to be distributed according to an exponential family. This also includes non-numeric (categorical) data such as binary labels.
> >
> > There should be more discussion about how restrictive your assumptions are in practice. What are  relevant application domains that generate data which actually conform to your modelling assumptions?

---

> > > ### Author Response · Authors · 2022-12-04
> > > **Responses to Reviewer fzvN (1/2)**
> > >
> > >
> > > **Q1**: Relationship to Network Lasso methods.
> > >
> > > **A**: Thanks for the suggestion. We will discuss the relationship in the revised paper.
> > >
> > > Before we answer the question, we first review the research problem focused by [3].
> > >
> > > > **Problem:** Given a graph $G$, assume that we observe
> > > > - node feature $x_i$ for each node i (i.e., content information) ,
> > > > - the labels of the nodes in a subset $\Omega$ (i.e., partially labelled data),
> > > >
> > > > **Goal:** The goal is learn node-wise classifiers which map node feature $x_i$ to the predicted label $\hat{y}_i$.
> > >
> > > To understand how [3] works, let us consider the binary classfication problem, and [3] solves the problem as follows:
> > >
> > > > **Exponential Family Classifiers:**
> > > > $\hat{y}_i= sigmoid(w_i^T x_i)$
> > > >
> > > > **Total Variation:**
> > > > $\parallel w \parallel_{TV}=\sum_{(i,j)\in{E}} \parallel w_i - w_j \parallel$, where $E$ is the set of edges
> > > >
> > > > **Objective:**
> > > > $\quad\quad \min_{w} -\sum_{i\in\Omega} sigmoid(w_i^T x_i) + \lambda \parallel w \parallel_{TV}$
> > >
> > > **Does [3] work for binary classification?** Yes, we fully agree with reviewer **fzvN** that it works for binary classification, when content features are provided.
> > >
> > > **Does [3] works in our setting?** We are afraid not. This is because 1) there is no content features in our setting, and 2) we only observe positive-unlabeled data, while it seems that true negative samples are available in [3]. We are afraid that the methods in [3] also suffer from degenerate solutions.
> > >
> > >
> > > **Q2:** More discussions of Poincare Condition.
> > >
> > > **A:** Thanks for the advice, and we will add more discussions in the revised paper.
> > >
> > > Let us first recall the Poincare condition as follows:
> > > > **Poincare condition:** Given a vertex subset $S\subset{V}$, the Poincare inequality is defined as
> > > >
> > > > $\quad\quad \parallel \phi \parallel \le  \Lambda\parallel L\phi\parallel $ for any $\phi\in{L}_2(S)$
> > > >
> > > >where $L$ is the graph Laplacian, $\Lambda$ is a positve scalar and ${L}_2(S)$ represents the Hilbert space of functions that map from set $S$ to real values.
> > >
> > > We also restate the Lemma 3.6 in [6]
> > > > Suppose that for a set of vertices $S\subset{V}$, the following holds true:
> > > > - every point from $S$ is adjacent to a point from the boundary $bS$, the set of all vertices in $V$ which are not in $S$ but adjacent to a vertex in $S$;
> > > > - for every $v\in{S}$ there exists at least one adjacent point $u_v\in b{S}$ whose adjacency set intersects $S$ only over $v$;
> > > > - the number $\Lambda$ is finite;
> > > >
> > > > Then the set S admits the Poincare inequality.
> > >
> > > We study on the Netflix data and we found nearly 900 users rigidly satisfy the Lemma 3.6. We also note that if we know the true negative examples, there are nearly 150,000 users that satisfy the Lemma.
> > >
> > > [6] Sampling in paley-wiener spaces on combinatorial graphs. Transactions of the American Mathematical Society.
> > >
> > >
> > > **Q3:** More discussion of modelling assumptions.
> > >
> > > **A:** Thanks for the suggestion. We will emphasize the modelling assumptions in the revised version.
> > >
> > > Theorem 4 states that GS-IMC can achieve the accurate reconstruction if $C_n=R(\omega)\parallel \mathbf{y}\parallel$ is not growing with $n$ (#items). In modern recommender systems, there are billions of products on the platform, and users will only purchase very few of them. In such case, $C_n$ is contant for most of users. We believe that this assumption also holds true for social networks. Thanks for pointing out this insight.

---

> > > ### Author Response · Authors · 2022-12-05
> > > **Responses to Reviewer fzvN (2/2)**
> > >
> > >
> > > **Q3**: Further discussion about our theory results.
> > >
> > > **A**: We sincerely thank the reviewer for the nice suggestion. We provide more discussions as below to address your concern.
> > >
> > > ## 1. Theorem 3 provides a guideline on how to sample negative data $\Omega_{-}$.
> > >
> > >
> > > Let us recall Theorem 3 here, where note that $\Omega=\Omega_+ \cup \Omega_{-} $: $\Omega_+$ represents the observed positive data, while $\Omega_{-}$ is the sampled negative data that is a mixture of unobserved positive examples and true negative examples.
> > >
> > > > **Theorem 3.** Given a regularization function $R(\lambda)$ with $\lambda\le{R}(\lambda)$ on graph $G=(V,E)$,
> > > > assume that $\Omega^c=V-\Omega$ admits the Poincare condition with a positive constant $\Lambda$,
> > > > then for any bandlimited function $f\in{PW}_\omega(G)$ with $0<\omega\le{R}(\omega)< 1/\Lambda$,
> > > >
> > > > $\quad\quad \parallel y - \hat{y}_k\parallel \le 2 ( \Lambda R(\omega) )^k \parallel y \parallel$
> > > >
> > > > and
> > > >
> > > > $\quad\quad \lim_{k\to\infty}  \hat{y}_k = y$
> > > >
> > > > where $k$ is a hyperparameter and $\hat{y}_k$ is the solution  of Eq. (5) with $\epsilon=0$.
> > >
> > > It is shown in Theorem 3 that if we can find a negative data set $\Omega_{-}$ that makes $\Omega^c$ admits the Poincare condition, then we can make an accurate reconstruction for the given bandlimited signal $f\in{PW}_\omega(G)$.
> > >
> > > **How to find such $\Omega_{-}$?**
> > > Fortunately, Lemma 3.6 in [6] answers the question to some extent. Let us consider a special case: if the vertices in $\Omega^c=V-\Omega$ are not connected to each other, then $\Omega^c$ satisfies all conditions in Lemma 3.6.
> > > > To achieve this, a heuristic is to reduce the degree of vertices in set $\Omega^c$, i.e., $d(\Omega^c)=\sum_{i\in\Omega^c} d(i)$ where $d(i)$ is the degree of the vertex $i$. If $d(\Omega^c)=0$, then vertices in $\Omega^c$ are disconnected.
> > >
> > > To do so, a greedy algorithm is to view the unlabeled data on high-degreed nodes as negative data. In the context of recommder systems, it is equivalent to treating unrated popular items as negative samples to the user. This is consistent with the weighting strategies in typical one-class collaborative filtering [7,8], which assumes that a miss on a popular item is more probable to be truly irrelevant to the user. Meanwhile, the successful application of the weighting strategies in [7,8] indicates that Theorem. 3 can be used to design new negative sampling strategies. We believe that this is valuable for the works on collaborative filtering.
> > >
> > > We thanks reviewer **fzvN** again for helping us improve the quality of our work!
> > >
> > > [7] One-Class Collaborative Filtering. ICDM 2008.
> > >
> > > [8] Fast matrix factorization for online recommendation with implicit feedback. SIGIR 2016.
> > >
> > >
> > >
> > > ## 2. Theorem 4 analyzes the reconstruction error of GS-IMC given a sampled $\Omega_{-}$.
> > >
> > > Despite that Theorem 3 provides the guideline on how to sample negative data, it has the following drawbacks when we use Theorem 3 to evaluate the reconstruction error:
> > > > 1. It ignores the discrete flip noise $\xi$ in $\Omega_{-}$. We recall that the unlabelled data is a mixture of unobserved positive examples and true negative examples. Hence, the sampled negative data introduces random noise to the model.
> > > > 2. The Poincare condition is restrictive. This assumption might not hold in some cases, which limits the applicability of Theorem 3.
> > >
> > > To address above issues, we derive Theorem 4 that considers the flip noise without the Poincare condition.
> > > > **Theorem 4.** Suppose that $\xi$ is the random noise with flip rate $\rho$,
> > > > and positive $\lambda_1\le\dots\lambda_n$ are eigenvalues of the graph Laplacian $L$,
> > > > then for any bandlimited function $y\in{PW}_\omega(G)$,
> > > >
> > > > $\quad\quad \mathbb{E} [MSE(y, \hat{y})] \le \frac{C_n^2}{n} ( \frac{\rho}{R(\lambda_1)(1+R(\lambda_1)/\varphi)^2}  + \frac{1}{4\varphi}  )$
> > > >
> > > > where $C_n = R(\omega) \parallel y\parallel$, $R(\Omega)$ is the regularization function, $\varphi$ is a hyperparameter and $\hat{y}$ is defined in Eq. (7).
> > >
> > > We see that Theorem 4 has no additional assumptions and takes into account random noise $\xi$. Hence, we believe that Theorem 4 is applicable for various real-world applications, such as social networks and recommender systems. Also, Theorem 4 provides many interesting results:
> > > > 1. For a constant $C_n=C>0$, our GS-IMC model can achieve accurate reconstruction;
> > > > 2. The error decreases with $R(\omega)$ declining. This observation indicates that low-frequency signal can be recovered more easily than high-frequency signal.
> > > > 3. When the flip rate $\rho\le 1/8$ is low, it is benefical to better fit the sampled negative data. However, when $\rho>1/8$ is high, we need to carefully tune the hyperparameter $\varphi$, and the optimal $\varphi*$ satisfies
> > > >
> > > > $ \quad\quad 1 + R(\lambda_1)/{\varphi*} = 2\rho^{1/3}$.
> > >
> > > We believe that our results to the 1-bit IMC problem are new in literature.
> > >
> > > We thank reviewer **fzvN** one more time for the nice suggestion, and we will adjust it in the revision.

---

> ### Author Response · Authors · 2022-11-19
> **Additional experiments**
>
>
> We finally achieve the results on Netflix of total variational (TV) approach which solves the following problem in [1]:
>
> $\min_f \frac{1}{2}\parallel s - f \parallel^2_2 + \lambda \parallel Lf \parallel_1 $
>
> where $s$ denotes the observations, and $L$ is the hypergraph Laplacian.
>
> | $\lambda$    | HR@10 | HR@50 | HR@100 |  NDCG@10 |  NDCG@50 |  NDCG@100 |
> | ------------- |:-------------:|:-------------:|:-------------:|:-------------:|:-------------:|:-------------:|
> | 1e-1      | $0.00028 \pm 0.0000$ | $0.00110 \pm 0.0002$ | $0.00226 \pm 0.0003$ | $0.00010 \pm 0.0000$ | $0.00027 \pm 0.0000$ | $0.00046 \pm 0.0000$
> | 1e-2      | $0.00028 \pm 0.0000$ | $0.00110 \pm 0.0002$ | $0.00226 \pm 0.0003$ | $0.00010 \pm 0.0000$ | $0.00027 \pm 0.0000$ | $0.00046 \pm 0.0000$
> | 1e-3      | $0.00761 \pm 0.0003$ | $0.01053 \pm 0.0001$ | $0.01328 \pm 0.0001$ | $0.00527 \pm 0.0002$ | $0.00589 \pm 0.0002$ | $0.00633 \pm 0.0002$
> | 1e-4      | $\mathbf{0.03829 \pm 0.0003}$ | $\mathbf{0.13020 \pm 0.0002}$ | $\mathbf{0.20313 \pm 0.0003}$ | $\mathbf{0.02193 \pm 0.0002}$ | $\mathbf{0.04133 \pm 0.0001}$ | $\mathbf{0.05310 \pm 0.0001}$
> | 1e-5      | $0.02523 \pm 0.0001$ | $0.09390 \pm 0.0011$ | $0.16076 \pm 0.0029$ | $0.01256 \pm 0.0001$ | $0.02706 \pm 0.0001$ | $0.03785 \pm 0.0004$
> | 1e-6      | $0.01956 \pm 0.0004$ | $0.07623 \pm 0.0003$ | $0.13370 \pm 0.0014$ | $0.00931 \pm 0.0003$ | $0.02123 \pm 0.0001$ | $0.03049 \pm 0.0004$
> | 1e-7      | $0.00028 \pm 0.0000$ | $0.00110 \pm 0.0002$ | $0.00226 \pm 0.0003$ | $0.00010 \pm 0.0000$ | $0.00027 \pm 0.0000$ | $0.00046 \pm 0.0000$
>
> To make a comparison, we recall the results of our GS-IMC and BGS-IMC using random walk regularization.
>
> |    | HR@10 | HR@50 | HR@100 |  NDCG@10 |  NDCG@50 |  NDCG@100 |
> | ------------- |:-------------:|:-------------:|:-------------:|:-------------:|:-------------:|:-------------:|
> | GS-IMC    | $0.09660 \pm 0.0006$ | $0.22328 \pm 0.0002$ | $0.32235 \pm 0.0011$ | $0.05452 \pm 0.0004$ | $0.08158 \pm 0.0004$ | $0.09759 \pm 0.0002$
> | BGS-IMC   | $0.09988 \pm 0.0006$ | $0.23390 \pm 0.0005$ | $0.33063 \pm 0.0009$ | $0.05593 \pm 0.0004$ | $0.08400 \pm 0.0004$ | $0.09982 \pm 0.0001$
>
> In a nutshell, these empirial results show that our proposed GS-IMC and BGS-IMC approaches perform better than typical TV approach on the largest Netflix data set.

---

> ### Author Response · Authors · 2022-11-26
> **Sincerely expecting further discussions from Reviewer fzvN**
>
> Dear Reviewer **fzvN**,
>
> We genuinely thank reviewer **fzvN** for your time. We hope our previous responses have addressed your concerns.
>
> As the discussion period is approaching its end, we would really appreciate it if you could kindly let us know whether there are any further questions. We will be more than happy to address them fully.
>
> Your Sincerely,
>
> Authors

---

> ### Author Response · Authors · 2022-12-03
> **Sincerely expecting further discussions from Reviewer fzvN**
>
> Dear Reviewer **fzvN**,
>
> We thank reviewer **fzvN** for the time of reviewing and the constructive comments. We really hope to have a further discussion with reviewer **fzvN** to see if our response solves the concerns.
>
> Here is a summary of our response.
>
> - We have added new figures to motivate the research problem and increase the readability.
>
> - We have added more experiment results of total variational approach.
>
> - We have revised the paper to emphasize the novelty and our contributions.
>
> - We have clarified the sampling theorem in Definition 3.
>
> - We have provided more explanations and insights for BGS-IMC.
>
> - We have provided more discussions on Theorem 3 (Appendix G.1) and Theorem 4.
>
> - We have provided more descriptions about the reproducibility in Appendix J.3 – J.4.
>
> We genuinely hope reviewer **fzvN** could kindly check our response. Thank you!
>
> Best Regards,
>
> Authors

---

> ### Author Response · Authors · 2022-12-09
> **Last three days reminder**
>
> Dear Reviewer **fzvN**,
>
> Many thanks for the additional feedback, especially for the questions about our theory results, which helps us to further enrich and improve our paper.
>
> To address all the concerns, we have provided point-wise responses.  Here is a summary of our responses.
>
> 1. We provided a toy example to motivate our research problem and describe our GS-IMC framework,
> > and note that existing SGMC and MRFCF are special cases of our framework.
> 2. We provided the computational complexity of GS-IMC, i.e., $O(lnk+k^3)$ linear to #nodes,
> > and note that our model achieves better results while consuming less training time.
> 3. We provided more discussions about the differene between the related work of [3] and GS-IMC,
> > and note that methods in [3] do not support our settings where there are no content features.
> 4. We provided more discussions of our theorems:
>    * Theorem 3 provides a guideline on how to sample negative data $\Omega_{-}$.
>    * Theorem 4 analyzes the reconstruction error of GS-IMC given a sampled $\Omega_{-}$.
>    * We only assume that the signal residing on graph nodes is bandlimited,
> > and note that GCN, GAT, ChebyNet, etc. are low-pass filters, and essentially they are also based on this assumption.
>
> We genuinely expect Reviewer **fzvN** could kindly check our responses and see if they successfully addressed your concerns. Thank you!
>
> Best Regards,
>
> Authors

---

### Official Review · Reviewer_rJLs · 2022-10-25

**Confidence:** 1
**Correctness:** 2
**Technical Novelty And Significance:** 2
**Empirical Novelty And Significance:** 2
**Recommendation:** 3

**Clarity, Quality, Novelty And Reproducibility:**

It is hard to follow even the first paragraph. This is hurting the readability of the whole paper.

- Why M has an extra column? What is the relationship between M and y?

- Since s is just a noisy version of y by flipping digits, is it necessary to define \xi in this manner, instead os saying something simple as "s_i = y_i with probability 1-\rho, and 1 - y_i with probability \rho"? As a reference, [1] defines the same mechanism in a much clearer way.

- Despite of saying "It is obvious now that the problem of inductive 1-bit matrix completion is equivalent to recovering clean y from corrupted s", I don't see the formal definition of the problem. What is the relationship between y and M, R, \Phi? It should be self-contained for readers.

References:
- [1] Davenport, Mark A., et al. "1-bit matrix completion." Information and Inference: A Journal of the IMA 3.3 (2014): 189-223.

**Strength And Weaknesses:**

Due to either: the lack of my knowledge in this area, or: the writing of the paper, I cannot understand what problem this paper tries to solve. I understand what one-bit matrix completion is, but I do not understand, at least from the first three Sections, how it is "unified" with graph signals or graph Laplacians. This is reflected in my confidence rating.

See Clarity, Quality, Novelty And Reproducibility for details.

- How should readers interpret the conditions in Theorem 3 and 4? Can the authors provide some examples in which they hold? Do these conditions hold in the experiments?

**Summary Of The Paper:**

The paper studies the problem of one-bit matrix completion. The paper claims to propose a unified graph signal sampling framework that enjoys the benefits of graph signal analysis and processing. The authors provide some theorems related to the quality of reconstruction.

**Summary Of The Review:**

The paper is hard to understand in the current shape.

---

> ### Author Response · Authors · 2022-11-13
> **We are sorry for causing some unnecessary difficulty for reading**
>
> We apologize for the difficulties when reading our paper. In the revised paper, we have added more contents and figures to increase the readability and clarity, and we expect the reviewer to kindly give us more comments.
>
> **Q**: Notations.
>
> **A**: We have added a new figure in the section to describe the difference between regular matrix completion (focuses on recovering missing values in $R$) and inductive matrix completion (aims to reconstruct unseen column $y$, and $M=[R, y]$). This figure also provides an example to understand the natural relationship between inductive matrix completion and graph signal sampling.
>
> Regarding to $\xi$, we agree that it is clearer that $s_i = y_i$ with probability $1-\rho$, and $y_i-1$ with probability \rho. The reason why we explicitly define the noise as $\xi\in$ {0, -1} is that we would like to emphasize the difference between our work and existing methods for total variation denoising.
>
> **Q**: Conditions in Theorem 3 and 4.
>
> **A**: We have added **Appendix G.1** to discuss the conditions in Theorem 3. As shown in Fig. 1, low-frequency signal represents user preference on popular items, while high-frequency signal reflects user preference on cold items (i.e., low degree vertices). From this viewpoint, when $R(\omega)$ is not sufficiently large, Poincare condition will not hold for the users who prefer niche products. This also explains why our GS-IMC model performs poorly for cold items (see Fig. 1, the recall for cold items is as low as 0.03, while the recall for popular items is up to 0.38). However, we argue that large $R(\omega)$ leads to improved accuracy while requiring more computational resources. There is a trade-off between the model accuracy and the system efficiency.
>
> Theorem 4 shows that accurate reconstruction can be made when $C_n$ is a constant. We believe that in context of recommender system this condition holds, since it is not possible for a user to buy all items in the platform.

---

### Official Review · Reviewer_idfJ · 2022-10-27

**Confidence:** 4
**Correctness:** 4
**Technical Novelty And Significance:** 3
**Empirical Novelty And Significance:** 3
**Recommendation:** 6

**Clarity, Quality, Novelty And Reproducibility:**

- The paper is written moderately clearly. To a person with less experience in related areas, this paper could be very difficult to follow, given their very condensed style of presentation. I gave suggestions to the authors to improve their exposition.

- the idea is original, in the sense that there is nothing ground-breaking that has been proposed, but the authors have managed to combine lots of different existing ideas from different sub-fields to come up with something reasonably new.

**Strength And Weaknesses:**

Strengths:

- The study cited many of the related prior art in a very comprehensive manner
- The core idea of the graph signal sampling approach + regularization is very natural, and admit simple closed form solutions
- reasonable error analysis is given for the method
- extensive experiments, with good results obtained for the proposed method against competitors


Weaknesses:

- the paper jumps directly into describing the problem of inductive 1-bit matrix completion. The motivation is insufficient in my opinion given that this is a very specific problem. I suggest that the authors spend a paragraph at the beginning talking about WHY the 1-bit formulation is helpful and why inductive matrix completion is useful so a broader audience can be reached.

- The "Bayesian" formulation, is not described clearly and sufficiently. The authors jump straight to describing a stochastic filtering problem and their prediction correction algorithm. There is minimal description of notation, and there is 0 discussion of model choice and why this model is sensible and as well as the choice of parameters such as Sigma_nu and Sigma_eta. For those who are not familiar with this model/literature, it can be very confusing what the prior is and why the proposed filtering algorithm works.

- Without more significant exposition for the Bayesian model, the current Bayesian model reads more like a distraction from the main theme of the paper.

I think the paper's readability and clarity would be greatly improved if the authors address the two issues above.


Minor Typos:

I think in the first paragraph of the first page, "a subset of positive examples phi randomly sampled from {(i, j) | ... .}" I think inside the set, in addition to j \leq m, there should also be i \leq n



**Summary Of The Paper:**

The authors propose a graph signal sampling approach to matrix completion/recommendation systems. They propose regularization approaches for noise reduction, and also provide a Bayesian extension that takes into account model uncertainty. They show that their approaches are scalable and provide both theoretical guarantees as well as experimental evaluations.

**Summary Of The Review:**

Overall, the authors proposed a reasonable and novel method for inductive 1-bit matrix completion using ideas from graph signal sampling. The main issue currently is their exposition. It could be a good paper if the authors improve their exposition, especially on their motivation of the inductive 1-bit matrix completion problem as well as their description of their "Bayesian" model.

---

> ### Author Response · Authors · 2022-11-13
> **Thanks for your nice suggestions**
>
> We quite enjoy reading your comments and really appreciate your nice suggestions that significantly improve the readability and clarity.
>
> We follow your advice step by step to revise our paper, where we have added more contents and figures to describe the motivation of inductive 1-bit matrix completion at the beginning of the introduction; we have also added figures in BGS-IMC section to describe how stochastic filtering problem is derived and the rationale behind the equations.
>
> **Q**: The property of $\Sigma_\nu, \Sigma_\eta$.
>
> **A**: This is a very interesting scientific inquiry.
>
> On one hand, $\nu$ represents the measurement noise which implicitly captures the uncertainty in observed positive data or ones, one of the advantages over GS-IMC that only considers the flip noise in unlabeled data or zeros. Typically, larger determinant of $\Sigma_\nu$ implies that data points are more dispersed, and BGS-IMC will lower the importance of measurements when the data uncertainty is high.
>
> On the other hand, $\eta$ models the process noise (e.g., the uncertainty of our model). It is interesting to see that when the determinant of $\Sigma_\eta$ is large, the Kalman gain is converged to identity matrix such that $\hat{x}_\mathrm{new}=z_\mathrm{new}$, namely BGS-IMC is reduced to GS-IMC. From this viewpoint, $\eta$ determines the difference between GS-IMC and BGS-IMC. We have added this observation in the revised paper. Thank you so much.

---

> ### Author Response · Authors · 2022-12-12
> **A thank you to our reviewer**
>
> Dear Reviwer idfJ,
>
> We would like to take this last opportunity to thank you for the time, valuable feedback, and nice recommendations for improvements. We know how much time and effort goes to writing a good peer review, and we deeply value the insightful guidance.
>
> It is not always easy to have a good peer review, and thus it is important for us to learn from your comments. The condensed writing style is not friendly to a reader who is not an expert in related domain. We have added new figures and contents to motivate our research problem and proposed algorithms in the revision. Thanks again for your nice comments on our article.
>
> Once again, many thanks for your time and support!
>
> Sincerely,
>
> Authors

---

### Author Response · Authors · 2022-11-13
**General Response for all Reviewers**

Dear Reviewers,

We thank all the reviewers for their constructive feedback and suggestions, **special thanks go to Reviewer idfJ**.

Based on this feedback we have added a substantial amount of new contents that we believe improved the quality of our paper. The main additions are summarized below.

- We have added three paragraphs and one figure to the beginning of **introduction section**, where we take recommender system as an example to motivate the problem of both regular and inductive (1-bit) matrix completion (Reviewer idfJ and rJLs), and also to clarify the definition of  ${M}, {R}, {y}, {s}$ as well as the noise $\xi$  (Reviewer ZrCo).

- We have added three paragraphs and one figure to **BGS-IMC section**, where we describe the motivation and model choice of BGS-IMC and also clarify the definition and property of  $\Sigma_\nu, \Sigma_\eta$   (Reviewer idfJ and ZrCo).

- We have added **Appendix G.1**, where we discuss more about the choice of $k$ and Poincare condition (Reviewer fzvN and rJLs).

To the end, we appreciate again that the reviewers considered our paper is novel and technically sound, provides comprehensive literature review, reasonable error analysis and extensive thorough experiments.

---

### Decision · Program_Chairs · 2023-01-20

**Decision:**

Accept: poster

**Justification For Why Not Higher Score:**

The approach applies GSP to inductive one-bit matrix completion. The methodology is sound and useful, but not novel enough to justify spotlight.

**Justification For Why Not Lower Score:**

The methodology is sound and useful, comes with theoretical guarantees, and performs favorably compared to baselines.

**Metareview: Summary, Strengths And Weaknesses:**

The paper attacks the inductive one-bit matrix completion problem using a graph signal processing perspective, and presents algorithms for the offline and online settings. Both algorithms have closed form solutions, are scalable, and come with theoretical guarantees. The method compares favorably experimentally against competitors.

**Note From Pc:**

if the above contains the word "oral" or "spotlight" please see: "oral" presentation means -> notable-top-5% and "spotlight" means -> notable-top-25%. As stated in our emails, we are disassociating presentation type from AC recommendations